

# On giant shoulders: How a seamount affects the microbial community composition of seawater and sponges

Kathrin Busch[1], Ulrike Hanz[2], Furu Mienis[2], Benjamin Müller[3], Andre Franke[4], Emyr Martyn Roberts[5], Hans Tore Rapp[5], Ute Hentschel[1,6]

[1]GEOMAR Helmholtz Centre for Ocean Research Kiel, Düsternbrooker Weg 20, 24105 Kiel, Germany
[2]NIOZ Royal Netherlands Institute for Sea Research and Utrecht University, 1790 AB Den Burg, Texel, The Netherlands
[3]University of Amsterdam, Science Park 904, P.O. Box 94248, Amsterdam, The Netherlands
[4]Institute of Clinical Molecular Biology (IKMB), Rosalind-Franklin-Straße 12, 24105 Kiel, Germany
[5]University of Bergen, Department of Biological Sciences and K.G. Jebsen Centre for Deep-sea Research, P.O. Box 7803, 5020 Bergen, Norway
[6]Christian-Albrechts University of Kiel, Düsternbrooker Weg 20, 24105 Kiel, Germany

*Correspondence to*: Ute Hentschel (uhentschel@geomar.de)

**Abstract.** Seamounts represent ideal systems to study the influence and interdependency of environmental gradients at a single geographic location. These topographic features represent a prominent habitat for various forms of life, including microbiota and macrobiota, spanning benthic as well as pelagic organisms. While it is known that seamounts are globally abundant structures, it still remains unclear how and to which extend the complexity of the seafloor is intertwined with the local oceanographic mosaic, biogeochemistry and microbiology of a seamount ecosystem. Along these lines, the present study aimed to explore whether and to which extend seamounts can have an imprint on the microbial community composition of seawater and of sessile benthic invertebrates, sponges. For our high-resolution sampling approach of microbial diversity (16S rRNA gene Amplicon sequencing) along with measurements of inorganic nutrients and other biogeochemical parameters, we focused on the Schulz Bank seamount ecosystem, a sponge ground ecosystem which is located on the Arctic Mid-Ocean Ridge. Seawater samples were collected at two sampling depths (mid-water: MW, and near-bed water: BW) from a total of 19 sampling sites. With a clustering approach we defined microbial micro-habitats within the pelagic realm at Schulz Bank, which were mapped onto the seamount's topography, and related to various environmental parameters (such as suspended particulate matter (SPM), dissolved inorganic carbon (DIC), silicate ($SiO_4^-$), phosphate ($PO_4^{3-}$), ammonia ($NH_4^+$), nitrate ($NO_3^{2-}$), nitrite ($NO_2^-$), depth, and dissolved oxygen ($O_2$)). The results of our study reveal a 'seamount effect' (sensu stricto) on the microbial mid-water pelagic community up to approximately 200 m above the seafloor. Further, we observed a strong spatial heterogeneity in the pelagic microbial landscape across the seamount, with planktonic microbial communities reflecting oscillatory and circulatory water movements, as well as processes of bentho-pelagic coupling. Depth, $NO_3^{2-}$, $SiO_4^-$, and $O_2$ concentrations differed significantly between the determined pelagic microbial clusters close to the seafloor (BW), suggesting that these parameters were presumably linked to changes in microbial community structures. Secondly, we assessed the associated microbial community compositions of three sponge species along a depth gradient of the seamount. While sponge-



associated microbial communities were found to be mainly species-specific, we also detected significant intra-specific

differences between individuals, depending on the pelagic near-bed cluster they originated from. The variable microbial phyla (i.e. phyla which showed significant differences across varying depth, $NO_3^{2-}$, $SiO_4^-$, $O_2$ concentrations and different from local seawater communities) were distinct for every sponge-species when considering average abundances per species. Variable microbial phyla included representatives of both, those taxa traditionally counted to the variable community fraction, as well as taxa counted traditionally to the core community fraction. Microbial co-occurrence patterns for the three examined sponge

species *Geodia hentscheli* (demosponge, HMA), *Lissodendoryx complicata* (demosponge, most likely LMA), and *Schaudinnia rosea* (Hexactinellida, most likely LMA) were distinct from each other. Over all, this study shows that topographic structures such as the Schulz Bank seamount can have an imprint ('seamount effect' sensu lato) on both, the microbial community composition of seawater and of sessile benthic invertebrates such as sponges by an interplay between the geology, physical oceanography, biogeochemistry and microbiology of seamounts.

## 1 Introduction


Seamounts and mid-ocean ridges are prominent geologic features that add to the complexity of the seafloor. There may be up to 100.000 to > 25 million seamounts present in the oceans (IUCN, 2013), although the error rate associated with these estimations is very high (IUCN, 2013). Despite the lack of accurate numbers, there is no doubt that with an estimated 10 million km² coverage, the area occupied by these habitats is globally significant. Elevated topographic features in the open

ocean are often hotspots of biological diversity and productivity (IUCN, 2013; Morato et al., 2010). It appears that interaction of the topography with the hydrography creates a combination of amplified tidal flow, increased current speed, and the formation of internal waves, which strongly enhances vertical mixing around seamounts (Lavelle and Mohn, 2010; Van Haren *et al.*, 2017; Roberts *et al.*, 2018). Consequential upwelling of nutrient-rich deep waters stimulates primary productivity in this layer of enhanced mixing. In addition to vertical mixing processes, also horizontal fluxes of organic matter may be affected

by the presence of seamounts, as they may promote enclosed or semi-enclosed oceanographic circulation patterns, like Taylor caps or columns [(Chapman and Haidvogel, 1992; Roberts et al., 2018) and references therein], leading to a retention of organic and inorganic matter.

Above mentioned processes make seamounts important habitats for pelagic as well as benthic species (Morato et al., 2010; Rogers, 2018) due to beneficial prevailing conditions. Particularly areas with strong water flows (evoked by interactions

of currents and tides with elevated topography), in combination with a steep and irregular hard substrate, represent suitable habitats for benthic suspension feeders, which indeed densely populate most seamounts (Genin et al., 1986; IUCN, 2013). Sponges (Porifera (Grant, 1836)) often dominate these suspension feeder communities and are increasingly recognised as key components of shallow and deep marine ecosystems (DeGoeij et al., 2017; Maldonado et al., 2016). Due to their high filtering capacity and association with diverse microbial communities, sponges are considered to substantially influence the carbon,

nitrogen, and silicate cycling in marine systems (Taylor *et al.*, 2007; Maldonado *et al.*, 2012; De Goeij *et al.*, 2013; Rix, De



Goeij, *et al.*, 2016; Maldonado *et al.*, 2019) and to contribute to benthopelagic coupling by actively removing particulate organic matter (POM) from the water column (Pile et al., 1997; Reiswig, 1971; Ribes et al., 1999). In addition to their influence on particulate organic matter pools, many sponges have been identified to primarily feed on dissolved organic matter (DOM) (De Goeij *et al.*, 2008; Mueller *et al.*, 2014; Hoer *et al.*, 2018; Gantt *et al.*, 2019). Energy and nutrients stored in this DOM are then transferred into particulate detritus, which fuels benthic food webs (De Goeij et al., 2013; Rix et al., 2016b).

Intimate sponge-microbe associations were observed throughout diverse habitats, reaching from coastal shallow sites in tropical and temperate regions to the deep-sea and polar seas (Helber et al., 2019; Kennedy et al., 2014; Moitinho-Silva et al., 2014; Naim et al., 2014; Schmitt et al., 2012; Steinert et al., 2019; Thomas et al., 2016). According to their microbiome, sponges can be classified to either feature high microbial abundance (HMA) or low microbial abundance (LMA) (Hentschel et al., 2003; Moitinho-Silva et al., 2017; Weisz et al., 2008). The dichotomy between HMA and LMA sponges is considered a main driver of the microbial community structure associated with shallow water sponges (Moitinho-Silva et al., 2017). In comparison to shallow waters, comparably few studies have been conducted on the microbiology of deep-sea sponges (Borchert et al., 2017; Jackson et al., 2013; Kennedy et al., 2014; Reveillaud et al., 2014; Tian et al., 2016). However, for example for deep-sea sponges of the genus *Geodia* (*G. barretti, G. macandrewii, G. phlegraei, G. atlantica*), similar microbial phyla have been observed as in HMA shallow water sponges, such as Acidobacteria, Poribacteria and Chloroflexi (Luter et al., 2017; Radax et al., 2012; Schöttner et al., 2013). In addition to the HMA-LMA dichotomy, an important factor in structuring the microbiomes of shallow water sponges is host taxonomy, which is manifested in ubiquitous species-specific sponge microbiomes (Easson and Thacker, 2014; Steinert et al., 2017; Thomas et al., 2016). Systematic analyses of the influence of biogeochemical parameters (particularly dissolved inorganic substances) on sponge-associated microbial diversity and interactivity is still lacking, particularly in deep-sea sponges. Seamounts provide an ideal study system in this regard, as they offer the potential of examining steep environmental gradients over small spatial scales. Sponge ground ecosystems are areas harbouring high densities of structure-forming sponge individuals. The Arctic Schulz Bank seamount has been observed to host a rich and diverse community of sponges (Roberts *et al.*, 2018; Meyer *et al.*, 2019) and may be considered a sponge ground ecosystem harbouring a reservoir of yet unexamined microbial biodiversity.

The present study aimed to characterise the microbial community composition of seawater surrounding the Schulz Bank seamount ecosystem, located on the Arctic Mid-Ocean Ridge. Seawater samples were collected at two sampling depths from a total of 19 sampling sites and the corresponding microbiome data were mapped onto the topography of the Schulz Bank seamount ecosystem. Secondly, we assessed the associated microbial community compositions of three sponge species along a depth gradient of the seamount. Diversity metrics, as well as changes in the abundance of individual microbial taxa were correlated with a set of biogeochemical parameters. This study explores whether topographic structures such as the Schulz Bank seamount, can have an imprint on both the microbial community composition of seawater and of sessile benthic invertebrates, sponges.





## 2 Methods

### 2.1 Description of the Schulz Bank seamount

Schulz Bank is located on the Arctic Mid-Ocean Ridge (73.8 °N; 7.5 °E) between the Greenland and Norwegian Seas. It is exposed to three main water masses: (i) the Norwegian Deep Water (NwDW) that is present at the base and flanks of the seamount, (ii) the intermediate water mass (NwArIW), which is most likely Norwegian Arctic Intermediate Water and occurs at the summit and shallower areas, and (iii) the warmer surface water mass (NwAtW) which is Norwegian Atlantic Water (Roberts et al., 2018). Notably near-bed water masses at Schulz Bank's summit have low temperatures of around 0 to -1 °C.

Estimations of the seamount's basal dimensions state minimal values of 10 x 4 km to 15 x 6 km (Roberts *et al.*, 2018). The summit of the seamount is located at around 600 m below the water surface. Bathymetry data presented in this study were derived from the Bathymetry Data Portal of the European Marine Observation and Data Network (EMODnet) and spatial analyses were performed in QGIS (version 3.4.4) as well as ArcGIS (version 10.6).

### 2.2 Sampling procedures

Three cruises were undertaken onboard RV *G.O. Sars* (campaign names 'GS2016109A', 'GS2017110', and 'GS2018108') during northern hemisphere's summer in the years 2016-2018. Seawater samples were collected with a rosette water sampler equipped with 12 x 10 L Niskin bottles combined with a CTD sensor system (SBE-9, Sea-Bird Electronics Inc., Washington, USA). In total, 19 CTD stations were covered, carried out along transects aligned with the seamounts' minor and major axes, and also with the 74 °N line of latitude. At each of the 19 stations, seawater samples for microbial analyses and biogeochemical

parameters were collected at two water depths during the CTD upcast: (i) 400 m below the seawater surface (mid-water) and (ii) correspondingly, at 10 m above the seafloor (near-bed water). Naturally, the near-bed depths varied along with seamount topography, which ranged from 575 to 2966 m. In particular, the near-bed water samples were collected at significantly different depths (ANOVA, p=0.01). Depth was lowest at the seamount's summit (average = 575 m), intermediate at the flanks (average ± SE = 919 ± 106 m and 922 ± 142 m, respectively), and greatest in the vicinity of the Schulz Bank seamount (average

± SE = 1836 ± 376 m).

Sponges were sampled between 580 and 2184 m water depth, along the CTD transects, by a remotely operated vehicle (ROV Ægir 6000, University of Bergen). A total of 36 sponge individuals were collected, including 16 *Geodia hentscheli* (Cárdenas et al., 2010) (Demospongiae), 8 *Lissodendoryx complicata* (Hansen, 1885) (Demospongiae)*,* and 12 *Schaudinnia rosea* (Fristedt, 1887) (Hexactinellida). The sponges were taxonomically identified by visual inspection on-board the ship. In

addition, whole specimens and additional sponge samples were fixed in 99 % EtOH for deposition in the collections of the University of Bergen.



## 2.3 Biogeochemical analyses and measurements of environmental parameters

The following nine environmental parameters were analysed: depth, suspended particulate matter (SPM), dissolved inorganic carbon (DIC), silicate (Si), phosphate ($PO_4^{3-}$), ammonia ($NH_4^+$), nitrate ($NO_3^{2-}$), nitrite ($NO_2^-$), and dissolved oxygen ($O_2$).

Depth and dissolved $O_2$ data were recorded during in situ water column profiling. Depth (pressure) was recorded with the CTD sensor system mentioned above. $O_2$ concentrations were derived from a dissolved oxygen sensor (SBE-43, Sea-Bird Electronics Inc., Washington, USA) that was attached to the rosette water sampler. For the analysis of suspended particulate matter (SPM), 2 x 10 L of water were filtered over pre-weighed combusted GFF filters, which were rinsed with demineralised water to remove salts (47 mm Whatman™ GF/F filters pre-combusted at 450 °C, stored at - 20 °C). Filters were freeze-dried and weighed before further analysis. For the analysis of inorganic nutrients (ammonia ($NH_4^+$), phosphate ($PO_4^{3-}$), nitrate ($NO_3^{2-}$), nitrite ($NO_2^-$), and silicate (Si)), seawater samples were filtered over 0.2 µm filters. Water samples for $NH_4^+$, $PO_4^{3-}$, $NO_x$ analysis were stored at - 20 °C and for Si analyses at 4 °C. Nutrients were measured with a QuAAtro Gas Segmented Continuous Flow Analyzer (Seal Analytical, Norderstedt, Germany). Measurements were made simultaneously on four channels for $PO_4^{3-}$ (Murphy and Riley, 1962), $NH_4^+$ (Helder and de Vries, 1979) and $NO_3^{2-}$ combined with $NO_2^-$ (Grasshoff et al., 2009) and separately for Si (Strickland and Parsons, 1972). A freshly diluted mixed nutrient standard containing Si, $PO_4^3$, and $NO_3^{2-}$ was added to each run. The cocktail served as a guide to monitor the performance of the standards. All measurements were calibrated with standards diluted in low nutrient seawater (LNSW). For the analysis of dissolved inorganic carbon (DIC), seawater samples were transferred into a glass vial containing 15 µL $HgCl_2$ (mercury chloride) and analysed on a TechniconTraacs800 auto-analyzer (Technicon Instruments Corporation, Tarrytown, USA) following the methodology of Stoll *et al.* (2001). Analyses of variance (ANOVAs) were performed to test for statistical differences in the biogeochemical and physical parameters between the determined microbial near-bed water clusters (see below).

## 2.4 Amplicon sequencing

Seawater samples were collected in triplicates from different Niskin bottles, yielding a total of 114 samples from all stations. Two litres of seawater sample were filtered onto polyvinylidene fluoride (PVDF) filter membranes (Merck Millipore) with a pore size of 0.22 µm and a diameter of 47 mm and stored at - 80 °C. Sponge samples were also collected at least in quadruplicates for each sampling region at Schulz Bank. Four cubes of approximately 1 cm³ were cut from the mesohyl with a scalpel, rinsed (sterile seawater), flash-frozen in liquid nitrogen and stored at - 80 °C. DNA was extracted from half a seawater filter or ~ 0.25 g of sponge tissue by using the DNeasy Power Soil Kit (Qiagen, Venlo, The Netherlands). The quality of the DNA extraction was assessed based on the 260/280 ratio using a NanoDrop spectrophotometer as well as by polymerase chain reaction with universal 16S primers and subsequent gel electrophoresis. The V3-V4 variable regions of the 16S rRNA gene were then amplified in a one-step PCR using the primer pair 341F-806R (dual-barcoding approach (Kozich et al., 2013); primer sequences: 5'-CCTACGGGAGGCAGCAG-3' & 5'-GGACTACHVGGGTWTCTAAT-3'). After verification of the presence of PCR-products by gel electrophoresis, normalisation (SequalPrep Normalisation Plate Kit; ThermoFisher Scientific,





Waltham, USA) and equimolar pooling was performed. Sequencing was conducted on the MiSeq platform (MiSeqFGx;
Illumina, San Diego, USA) with v3 chemistry. The settings for demultiplexing were 0 mismatches in the barcode sequences.

**2.5 Bioinformatic analyses**

For computation of microbial core-diversity metrics, sequences were processed within the QIIME2 environment (version
2018.11, (Bolyen et al., 2018)). Amplicon Sequence Variants (ASVs) were generated from forward reads (truncated to 270nt)
with the DADA2 algorithm (Callahan et al., 2016). Phylogenetic trees were calculated based on resulting ASVs with the
FastTree2 plugin. Representative ASVs were classified using the Silva 132 99 % OTUs 16S database (Quast et al., 2013) with
the help of a primer-specific trained Naive Bayes taxonomic classifier. Alpha and beta diversity indices (e.g. Faith's
Phylogenetic Diversity and weighted UniFrac distances, respectively) were calculated within QIIME2. To evaluate sample
separation in ordination space, non-metric multidimensional scaling (NMDS) was performed on weighted UniFrac distances
for seawater and sponge-associated microbiomes separately.

170          A machine learning approach was used to define microbial micro-habitats within the pelagic realm. Seawater
microbiomes were clustered based on weighted UniFrac distances. The NbClust function was applied in R (version 3.0.2, (R
Development Core Team, 2008)) to generate 30 indices to identify the best number of clusters based on the majority rule. A
coordinate grid was set up as a basis for a georeferenced extrapolation of sampling points. Clustering regions were set up with
the help of the k-Nearest-Neighbor-algorithm. The machine learning approach was fine-tuned in several ways: (i) the algorithm
was trained in a way that in situ measured data points always belong to the cluster actually determined based on the sequencing
data; (ii) a normalisation was applied with the help of a distance-weighted function meaning that closer data points have a
higher weight; (iii) the probability of class membership was calculated and plotted as indication of confidence. Permutational
multivariate analyses of variance (PERMANOVAs) were performed with 999 permutations to determine whether microbiomes
of selected clusters were statistically significantly different from each other. In detail, pair-wise tests across the determined
clusters were conducted for the following samples separately: mid-water samples, near-bed water samples, *G. hentscheli*, *L.
complicata*, and *S. rosea*. A significance level of $\alpha=0.05$ was applied for all statistical analyses in this study.

To evaluate co-occurrence patterns between microbial taxa across environmental gradients (i.e. determined near-bed
water clusters), networks were constructed separately for every sponge species and seawater. Mean relative abundances of
microbial phyla were calculated for all biological replicates of each sample type and for the corresponding near-bed water
cluster. Microbial phyla, which showed significantly different enrichment between clusters, were determined and ranked using
the Linear Discriminant Analysis Effect Size (LEfSe) algorithm (Segata et al., 2011). A correlation matrix was established for
those taxa that differed significantly between clusters, to assess co-occurrences. In particular, the direction and strength of
correlations were characterised for any significant phylum with all other significant taxa (as well as with depth).



## 3 Results

### 3.1 Structure and composition of seawater microbial communities

A NMDS plot on weighted UniFrac distances separated the microbial communities of mid-water and near-bed water samples in ordination space with few exceptions (Fig. 1). Cluster analysis based on weighted UniFrac distances revealed two distinct clusters in the mid-water samples of which one (MW1) was located precisely above the summit of Schulz Bank, while the other (MW2) covered the wider seamount region and vicinity (Fig. 2A). Four distinct microbiome clusters were detected in

the near-bed water samples (BW1-4). In terms of similarity, cluster BW1 was most distinct from all other clusters while clusters BW2 and BW3 were most similar to each other (Fig.2B). Moreover, BW1 cluster samples separated in ordination space in that they grouped with mid-water rather than near-bed water samples (i.e. consider the few black dots grouping together with the white dots in Fig. 1). Plotting the clusters on a spatial map revealed that near-bed water cluster BW1 was located near the summit of Schulz Bank seamount, while clusters BW2 and BW3 covered its flanks, and cluster BW4 represented the vicinity

close to the seamount (Fig.2 B). Fig. 2C shows the bathymetry highlighting the contour lines of Schulz Bank seamount and its vicinity (reference West and East) as well as the 19 sampling stations. In addition to this representation, a 3D visualization of the microbiome clusters at and around Schulz Bank seamount was created (Fig. 3). Here, a digital elevation model of Schulz Bank seamount is depicted in combination with the overlaying water column structure and oceanographic context. Temperature profiles derived from whole water column sensing by CTD casts are plotted. Based on these profiles the vertical distributions

of the surface water (NwAtW), intermediate water (NwArIW), and North Atlantic Deep Water (NwDW) were deduced in combination with the identified water masses as described in Roberts *et al.* (2018) (Fig. 3).

Microbial richness was overall slightly lower in the mid-water samples (mean Faith's Phylogenetic Diversity ± standard error = 45.5 ± 0.8) than in the near-bed water samples (54.4 ± 0.9) (Supplementary Material S1). Near-bed water samples from the summit (BW1) represented an exception to this pattern as they displayed a slightly lower microbial richness

than the other near-bed water samples. The mid-water samples collected above Schulz summit showed also a slightly lower microbial richness than the other mid-water samples. Pairwise comparisons (PERMANOVA) revealed that the seawater microbial community clusters within the mid-water and near-bed water samples were significantly different from each other (Supplementary Table S1). Furthermore, the pool of mid-water samples was significantly different from the pool of near-bed water samples. Overall, the eight most dominant seawater microbial phyla, sorted in descending order of mean relative

abundance, were: Proteobacteria (54 % of total community), Bacteroidetes (17 %), Verrucomicrobia (7 %), Marinimicrobia (SAR406 clade) (6 %), Actinobacteria (5 %), Chloroflexi (4 %), Acidobacteria (2 %), and Planctomycetes (1 %).

### 3.2 Seawater biogeochemistry at Schulz Bank seamount

When comparing the biogeochemical parameters of the mid-water samples, only dissolved $O_2$ concentrations differed significantly (ANOVA, p=0.02) with slightly higher concentrations in MW1 (6.90 ± 0.04 mL $L^{-1}$) compared to MW2 (6.75 ±

0.03 mL $L^{-1}$) (Supplementary Table S2). All other tested biogeochemical parameters (SPM (2.22 ± 1.39 mg $L^{-1}$), DIC (2269.07





$\pm$ 26.63 µmol L$^{-1}$), SiO$_4^-$ (5.66 $\pm$ 0.06 µmol L$^{-1}$), PO$_4^{3-}$ (0.86 $\pm$ 0.01 µmol L$^{-1}$), NH$_4^+$ (0.11 $\pm$ 0.03 µmol L$^{-1}$), NO$_3^-$ (12.99 $\pm$ 0.08 µmol L$^{-1}$), and NO$_2^-$ (0.02 $\pm$ 0.01 µmol L$^{-1}$) were not statistically different between MW1 and MW2. The values for mid-water samples are reported as average $\pm$ standard error.

225         Of the eight biogeochemical parameters tested, the following three differed significantly between the near-bed water clusters. These were NO$_3^-$ (ANOVA, p = 0.02), SiO$_4^-$ (ANOVA, p = 0.02), and dissolved O$_2$ (ANOVA, p = 0.01). Nitrate (range= 13.00-14.78 µmol L$^{-1}$) and SiO$_4^-$ (range = 6.00-10.65 µmol L$^{-1}$) increased with depths, with lowest concentrations at the summit (BW1), intermediate concentrations at the flanks (BW2, BW3) and highest concentrations in the seamount vicinity (BW4) (Fig. 4). Dissolved oxygen (range= 6.48-6.99 mL L$^{-1}$) showed the reverse pattern in that its concentration was highest at the summit (BW1), intermediate at the flanks (BW2, BW3) and lowest in the seamount vicinity sites (BW4). The other
biogeochemical paramenters SPM (range= 0.49-1.87 mg L$^{-1}$), DIC (range= 2248.00-2265.67 µmol L$^{-1}$), PO$_4^{3-}$ (range= 0.90-0.97 µmol L$^{-1}$), NH$_4^+$ (range= 0.10-0.17 µmol L$^{-1}$), and NO$_2^-$ (range= 0-0.05 µmol L$^{-1}$) were not significantly different between the near-bed water clusters. At the summit, no pronounced differences in biogeochemical parameters were observed between the near-bed water (BW1) and mid-water samples (MW1) (Supplementary Table S2).

### 3.3 Structure and composition of sponge microbial communities

Overall, the three deep-sea sponge species *S. rosea*, *G. hentscheli*, and *L. complicata* showed host species-specific microbiomes, as indicated by a clear separation of their microbial communities in ordination space (Fig. 5). Sub-structuring based on near-bed water clusters in the non-metric multidimensional scaling plot as well as pairwise comparisons (PERMANOVA) revealed that the sponge microbial communities within each species differed significantly depending on the near-bed water clusters from which they were collected (Supplementary Table S1). The only exception was *S. rosea*, for which
specimens from the flank (BW3) showed a microbial community composition that was intermediate between the summit (BW1) and the other flank cluster (BW2).

        The dominant microbial phylum in *S. rosea* and *L. complicata* were Proteobacteria (Fig. 6A and Fig. 6B), whereas *G. hentscheli* microbiomes were dominated by Chloroflexi, Acidobacteria and Proteobacteria (Fig. 6C). Sponge microbiomes were more stable than seawater communities with less phyla exhibiting significant differences across the four near-bed water
clusters or positively correlating with depth (Fig. 6). For the hexactinellid *S. rosea*, the relative abundances of five bacterial phyla (Acidobacteria, Chlamydiae, Kirimatiellaota, Planctomyces and Proteobacteria) were significantly different between individuals that were sampled from different near-bed water clusters. Out of these five phyla, the Acidobacteria, Chlamydiae, Kirimatiellaota, and Planctomyces were positively correlated with depth while for the Proteobacteria neither a positive nor negative correlation with depth was discernable. Consequently, the Proteobacteria showed a negative correlation with the four
other phyla in the network analysis. For the demosponge *L. complicata*, the relative abundances of the Bacteroidetes, Gemmatimonadetes, Nitrospinae, Planctomyces, Proteobacteria and Spirochaetes were significantly different between sponge individuals that were sampled from the different near-bed water clusters. For this sponge species, samples were only available from near-bed water clusters 1 and 2. Of the six phyla, the Planctomyces and Proteobacteria were positively correlated with





depth, while the other four were negatively correlated with depth, which is also reflected in the network analysis. For the

demosponge *G. hentscheli*, the relative abundances of eight phyla (Acidobacteria, Actinobacteria, Bacteoidetes, Chloroflexi, Dadabacteria, Entotheonellota, PAUC34f and Schekmanbacteria) were significantly different between sponge individuals sampled from the near-bed water clusters BW1-BW4. Of those, Chloroflexi and Schekmanbacteria were positively correlated with depth, while the others showed variable patterns over depth. The network analysis showed both positive and negative correlations between taxa for those increasing with depth as well as those displaying a variable response.

When analysing host-associated microbiomes, ambient seawater microbiomes are valuable references for comparison. In this study, a total of 21 microbial clades were identified in ambient seawater, whose relative abundances varied significantly between the four near-bed water clusters. A total of nine taxa showed a positive correlation with depth, one (Dadabacteria) showed a negative correlation with depth, and the remaining 11 taxa showed a variable response to depth. For seawater more phyla varied between near-bed water clusters than for the sponge samples. Overall, more microbial taxa showed

significant positive correlations with depth, $NO_3^-$, $SiO_4^-$ and negative correlations with $O_2$ than vice versa. The microbial taxa showing a significant variability between near-bed water clusters were different ones between sponges and seawater, and also between the sponge species. Further, the pattern applied to both abundant and less abundant bacterial lineages.

## 4 Discussion

Research records about seamount microbiology are sparse and comparably few studies have been conducted on deep-sea

sponge microbiomes in general (Borchert et al., 2017; Jackson et al., 2013; Kennedy et al., 2014; Reveillaud et al., 2014). Our main aim was to assess whether and via which potential mechanisms a seamount can affect the community structure of pelagic and benthos (sponge)-associated microbial communities, using the Schulz Bank seamount as an exemplary field site. A total of 19 CTD sampling stations, each with two sampling depths, on and around Schulz Bank were analysed towards this goal, and combined with sponge-associated microbial data gained during additional ROV dives.

**4.1 A seamount imprint on seawater microbial communities**

In this study we observed a pronounced similarity between the microbial community composition of the mid-water cluster located precisely above Schulz Bank's summit (MW1) and the microbial community composition of the near-bed water cluster at the summit (BW1). In addition, the microbial community in the mid-water cluster above Schulz Bank's summit (MW1) was distinct from the community in the mid-water cluster covering the wider seamount region and vicinity (MW2), despite similar

prevailing biogeochemical conditions in both mid-water clusters (MW1 vs MW2; exception = significant difference in $O_2$ concentrations between both clusters). From these two observations we conclude that the presence of a seamount can have an imprint on the microbial community structure in the overlying water column ('seamount effect' sensu stricto). In particular we suspect that topography-induced vertical mixing processes occur at Schulz Bank seamount, which reallocate microbial communities within the water column and in turn influence the pelagic microbial diversity as far as approximately 200 m





above the seamount's summit. In support of these interpretations, oscillating currents relating to the barotropic and baroclinic (internal) tide have been reported previously at the summit of the Schulz Bank seamount (Roberts et al., 2018) and other seamounts (Van Haren et al., 2017).

In addition to tide-induced vertical hydrodynamic processes, horizontal flow patterns can also help to explain the presence of seamount-specific microbial communities. Roberts *et al.* (2018) calculated that a Taylor cap or Taylor column

may be (temporarily) present at Schulz Bank. This oceanographic phenomenon describes an isolated anti-cyclonic flow circulation pattern over a seamount in the Northern hemisphere and hence may promote temporal isolation of a seamount ecosystem from adjacent waters. Our conceptual schematic overview of such events at Schulz Bank seamount is shown in Fig. 3 (dashed lines). This Figure recapitulates that microbial signatures of seawater in this study were shown to be consistent with the oscillatory water movements (i.e., due to internal tide-induced mixing) and possible circulatory flows (Taylor column) as

predicted by Roberts *et al.* (2018).

Microbial richness was overall slightly lower at the summit of the seamount (BW1) than at the deeper locations (i.e. BW2-BW4). A similar trend was observed for the mid-water samples, where microbial richness was slightly lower for the microbial community above Schulz's summit in comparison to samples in its vicinity (MW1 vs MW2). On a macroscopic level, Morato *et al.* (2010) and others have described seamounts as hotspots of pelagic biodiversity. Our results of a lower

microbial richness above the seamount summit might seem contradictory at first. However, Schulz Bank is a recognized sponge ground ecosystem, with a peak in sponge density and diversity at the seamount summit (Roberts *et al.*, 2018; Meyer *et al.*, 2019). Sponges are very efficient suspension feeders and are known for removing large amounts of particulate organic matter including prokaryotes and small eukaryotes from the water column (Leys et al., 2018). Benthic-pelagic coupling mediated by selective feeding of sponges on seawater microorganisms (McMurray et al., 2016; Van Oevelen et al., 2018) in combination

with the discussed hydrodynamic patterns (vertical mixing) might explain the slightly reduced microbial richness of the water body residing directly above a sponge ground (BW1 and MW1). The circumstance that sponge density and community composition changes along the topography of Schulz Bank seamount (Meyer et al., 2019; Roberts et al., 2018), can further aid to explain observed differences in microbial community composition between the other near-bed water samples (BW2-4). For these samples, we observed distinct pelagic microbial communities at a finer resolution than can be explained by the pure

water masses distribution (consider depth of intermediate and deep water layers in Fig. 3). Particularly the near-bed clusters BW2 and BW3, originated from a similar depth range and both were located at the seamount's flanks. Besides ecologically rooted explanations (i.e. varying presence of dense ecosystems that influence biogeochemical cycles), also hydrodynamic processes (i.e. local flow direction linked to small scale topography and/or spatial orientation of the seamount's flanks) can explain the observation of distinct microbial community compositions within the near-bed water. We hence conclude that

besides patterns related to water masses, we observe a much higher spatial heterogeneity of pelagic microbial communities than previously recognized. We call this kind of imprint on the pelagic microbial community composition, which is based on the topography combined with bentho-pelagic coupling processes and hydrodynamics, a 'seamount effect sensu lato'. These





observations suggest that the presence of a seamount can have profound impacts on the distribution of microbial landscapes in the open ocean.

Seamounts are recognized as unique habitats in terms of ecosystem dynamics (Genin and Boehlert, 1985) and macroecology (Morato et al., 2010). The present study reveals that seamounts also have a unique microbial signature that extends hundreds of meters up into the overlying water column. In addition, we detected three distinct microbial clusters in seawater samples taken near the seabed directly above Schulz Bank which were distinctly different from those of seawater collected in the vicinity of the seamount (near-bed water clusters BW1-3 vs BW4). $NO_3^-$, $SiO_4^-$, and dissolved $O_2$ concentrations differed

significantly between the four near-bed water clusters. The observation, that the seamount is intersecting with different biogeochemical properties and microbial communities, is particularly interesting in regard to benthic organisms. In these lines, the Schulz Bank seamount provides a platform for sponges and their associated microbial communities to respond to topography-enabled environmental gradients (also a 'seamount effect sensu lato').

### 4.2 A seamount imprint on sponge-associated microbial communities

The investigated sponge species *S. rosea*, *G. hentscheli* and *L. complicata* were selected for this study as they represent key taxa of the sponge community at Schulz Bank. Microbiomes of these three species clustered clearly apart from each other in ordination space, indicating a dominant host species-effect on the associated microbial community structure. *S. rosea* and *L. complicata* showed characteristic microbial signatures of LMA sponges (as defined in Moitinho-Silva *et al.*, 2017) that are being dominated by Proteobacteria. On the contrary, *G. hentscheli* displayed a microbial signature characteristic of HMA

sponges with dominant clades such as Chloroflexi and Acidobacteria, which is consistent with previous reports on sponges of the genus *Geodia* (Luter et al., 2017; Radax et al., 2012; Schöttner et al., 2013).

When analysing each sponge species separately, sponge specimen microbiomes differed significantly between each other and depended on the near-bed water clusters to which they belonged. This finding suggests that an environmental signature is also detectable in sponge-associated microbial communities (seamount signature sensu lato). This observation is

striking, as sponge-associated microbial communities are considered as highly stable associations (Cárdenas et al., 2014; Erwin et al., 2012, 2015; Pita et al., 2013; Steinert et al., 2016).

Previous studies have shown that abiotic factors (i.e. depth, geographical location) influence the microbial community structure in shallow-water sponges, but stated that the core community is shaped by the intimate interaction with the sponge host (Lurgi et al., 2019). Interestingly, in our study the major microbial players in terms of abundance, such as Chloroflexi in

*G. hentscheli*, show significant enrichment/depletion patterns across the four clusters. Traditionally (shallow water) sponge-associated microbes have been classified into core, variable and species-specific communities (Schmitt et al., 2012). The present study reveals that for the three investigated deep-sea sponges at Schulz Bank seamount the variable community overlaps with the core-community when considering high taxonomy ranks.

In this study, silicate, oxygen and nitrate concentrations, as well as depth differed significantly between the four near-

bed water clusters. While $SiO_4^-$ and $NO_3^-$ were positively correlated with depth, $O_2$ showed a negative relationship. An increase





of nutrient concentrations with depth is consistent with our previous expectations [consider e.g. (Bristow et al., 2017)] and can be explained by remineralization processes of sinking marine snow within the deep water layers. Decreasing $O_2$ concentrations from the Intermediate Water (NwArIW) to the Deep Water (NwDW) are also consistent with our prediction based on physical oceanography, as water layers more recently oxygenated at the ocean surface at their site of formation, typically carry more
oxygen (Jeansson et al., 2017). In general, the absolute differences in the concentrations of all three significant environmental parameters were comparably small (especially $NO_3^-$ and $O_2$). However, as microbial communities were significantly different between the clusters, we posit that the observed biogeochemical differences, albeit small, should be considered as drivers of sponge microbial community composition. In support of our hypothesis, the process of denitrification is for example known to be highly sensitive to nanomolar concentrations of $O_2$ concentrations (Dalsgaard et al., 2013). In addition, a previous study
on the sponge *Xestospongia muta* demonstrated that changing NOx concentrations over depth contribute to shaping the microbial community composition (Morrow et al., 2016). Furthermore, several studies have noted that depth is an important factor in structuring sponge-associated microbiomes (Indraningrat et al., 2019; Lesser et al., 2019; Steinert et al., 2016).

In this study, the relative abundance of Chloroflexi (among several other phyla) differed significantly between the four near-bed water clusters for *G. hentscheli* and seawater, showing a positive correlation with $NO_3^-$, $SiO_4^-$, and depth, and a
negative correlation with $O_2$. Members of the phylum Chloroflexi have been attributed to a relevant role in the degradation of organic matter, particularly in the deep ocean pelagic realm and within HMA-sponges (Bayer et al., 2018; Landry et al., 2017). High degradation rates of organic matter are often related with low $O_2$ and high nutrient concentrations, owing to biogeochemical feedbacks where nutrients enhance oxygen demand by increasing biological production and oxygen consumption during decomposition. Taken together, the differences in relative abundances of Chloroflexi in *G. hentscheli* and
seawater could be driven by the $NO_3^-$, $SiO_4^-$, and $O_2$ concentrations in ambient seawater. While sponge microbiomes are generally considered as being highly stable in time and space, we provide a first evidence that small differences in water biogeochemistry may affect sponge microbiome composition. However, no uniform shifts in relative abundances of microbial taxa were observed for *G. hentscheli*, *L. complicata*, and *S. rosea*, but rather an individual response of each host species related to biogeochemical parameters. One explanation is that biological interactions between the sponge host and its microbiome, or
between the microbes themselves might have masking effects.

**Conclusions**

We provide insights into the variability of pelagic and benthic (sponge-associated) microbiomes at the Arctic Schulz Bank seamount with high resolution sampling. Interestingly, a 'seamount signature' is detected within the microbial community composition of samples originating as far as 200 m above the seamount summit. We further show that the biogeochemistry of
seawater which varies over depth ($NO_3^-$, $SiO_4^-$, and $O_2$ concentrations) has a detectable, but variable influence on the composition of sponge-associated microbiomes. This study provides new perspectives on the influence of seamounts on the



microbial diversity in their vicinity. We conclude that the geology, physical oceanography, biogeochemistry and microbiology of seamounts and similar structures are even more closely linked than currently appreciated.

## Data availability

Sample metadata and biogeochemical data were deposited in the Pangaea database: https://doi.pangaea.de/10.1594/PANGAEA.911304. Raw sequences were archived in the NCBI Sequence Read Archive under BioProject id: PRJNA600711.

## Author contribution

KB, FM, BM, UHe designed the study. KB, UHa, FM, EMR, HTR participated in sampling. UHa, FM, KB focused on
biogeochemical parameters. HTR conducted the sponge taxonomic analysis. UHe, KB were responsible for the microbial pipeline. AF was involved in sequencing. KB performed the data analysis (bioinformatics and visualisations). KB, UHe wrote the manuscript. BM, FM, EMR, UHa, HTR reviewed and edited the manuscript.

## Competing interests

The authors declare that they have no conflict of interest.

## Acknowledgements


This study was funded by the European Union's Horizon 2020 research and innovation program under Grant Agreement No. 679849 (the SponGES project). This document reflects only the authors' view and the Executive Agency for Small and Medium-sized Enterprises (EASME) is not responsible for any use that may be made of the information it contains.

Samples included in this study were collected in compliance with the Nagoya Protocol. We thank the crews and scientific
parties of RV *G.O. Sars* cruises 'GS2016109A', 'GS2017110', and 'GS2018108' for great technical support while at sea. We are grateful for sponge sampling by Stig Vågenes and his ROV Ægir 6000-team (UiB), Christine Rooks (UiB) for sampling assistance on-board ship during 'GS2016109A', and Jasper de Goeij (UvA) for interesting discussions. We further acknowledge Ina Clefsen, Andrea Hethke, Ilona Urbach and Tonio Hauptmann (Kiel, Germany) for excellent laboratory support with the amplicon pipeline. Corinna Bang (IKMB, Kiel) provided valuable support with the sample sequencing and
revised the manuscript before submission.





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





**Figure 1: Seawater microbial community composition of mid-water and near-bed water samples visualised by a non-metric multidimensional scaling plot on weighted UniFrac distances. Each marker is one microbial community, with colors indicating the sample sub-type (i.e. mid-water or near-bed water).**





**Figure 2: Seawater microbial community structure across Schulz Bank.** Contour lines in all three subplots represent the underlying topography. Colors in A) and B) represent clusters based on weighted UniFrac distances, where colored dots indicate stations with in situ sampling and filled areas represent extrapolations based on machine learning. A) includes all mid-water samples derived during the CTD transects. B) includes all near-bed water samples. Here, the degree of cluster similarity can be deduced from the dendrogram to the right of the plot(s). C) provides an overview of the sampling area, showing the locations of all 19 CTD stations. Stations directly located on the Schulz Bank are coloured yellow, while reference stations (west and east of Schulz Bank) are indicated by black colours. Colouring in sub-plot C) was done according to depth.



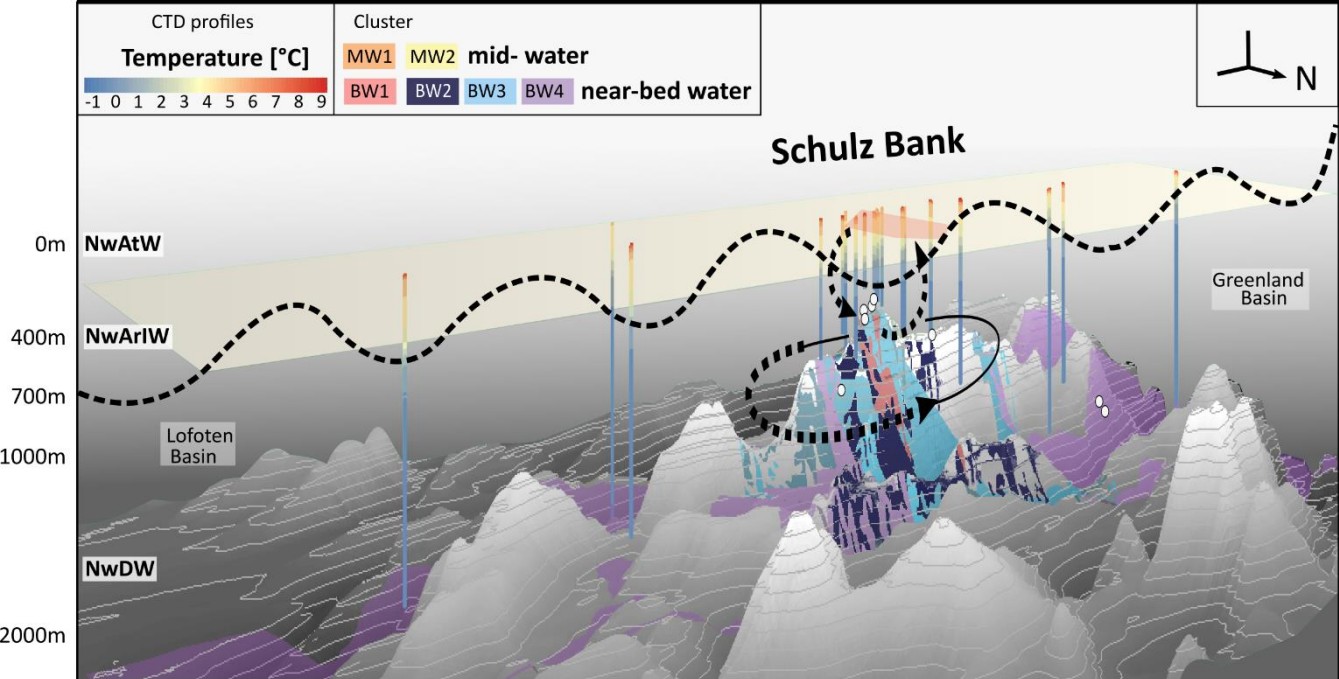

**Figure 3: Conceptual overview and vertical 3D section showing spatial distribution of microbial clusters and oceanographic patterns on the Schulz Bank seamount. Extrapolated seawater microbial clusters are indicated by colored polygons: mid-water clusters are marked in orange (MW1) and yellow (MW2), while near-bed water clusters are marked in red (BW1), dark blue (BW2), light blue (BW3), and purple (BW4). Whole water column CTD profiles are indicated, showing the measured temperature values from surface to bottom at the respective sampling locations. Sponge sampling locations visible on this side of the seamount are indicated by white balls. Vertical positions of major watermasses: Norwegian Atlantic Water (NwAtW), Norwegian Arctic Intermediate Water (NwArIW), and Norwegian Deep Water (NwDW) are indicated. To give a broad orientation in space, a north arrow is depicted, as well as the major geologic features (Lofoten Basin and Greenland Basin). For Schulz Bank, water flows, such as a potential Taylor column circulating around the seamount, mixing between summit and pelagic realm, as well as tidally-driven internal motions (black horizontal line) are indicated by dashed arrows and lines.**



**Figure 4: Concentrations and measurements of significant (ANOVA, α=0.05) biogeochemical parameters for near-bed water samples, across the determined near-bed water clusters. p-values for these parameters are written into the respective graphs. Colouring is the same as chosen for Fig. 2 and 3.**





**Figure 5: Sponge microbial community composition visualised by a non-metric multidimensional scaling plot on weighted UniFrac distances. Each marker is one microbial community, with symbols representing the sample sub-type (i.e. sponge species) and colors indicating the near-bed water cluster present at the respective sponge sampling location.**







**Figure 6: Co-occurrence network and differential abundance of microbial phyla across the four determined near-bed water clusters.** Sub-plots A-C) show sponge data, with plot A) showing average *Schaudinnia rosea* data, B) presenting average *Lissodendoryx complicata* data and C) illustrating *Geodia hentscheli* data. Sub-plot D) contains seawater data. Near-bed water clusters are represented by differently colored rings. Each ring contains a list with microbial phyla which are alphabetically sorted. Average relative abundances of each of the respective phyla for the samples within a given cluster are indicated by bubble sizes. Those microbial phyla which are statistically significantly enriched or depleted across the four clusters (LefSE analysis), are marked with an asterisk inside the inner most ring. Only for those taxa where the difference is significant, correlation strength (indicated by size of connecting lines) and direction (represented by color of connecting lines: white = negative correlation, dark grey = positive correlation) with all other significant taxa are plotted. For all microbial phyla correlation with depth is indicated in the outer ring of each plot by + (meaning significant positive correlation) or - (meaning significant negative correlation).