# Peer review of "On giant shoulders: How a seamount affects the microbial community composition of seawater and sponges"

_Biogeosciences, 2020_

## Referee Comment (RC1) · Anonymous Referee #1 · 31 Jan 2020

Overall comments:

This manuscript brings to bear physical oceanographic, biogeochemical, and microbiological data on the question of whether seamounts impact the microbial community composition of the water column and benthic organisms like sponges. I was skeptical of water column impacts, imagining that a given water mass would have a microbial signature irrespective of the seamount. However, the data in this paper convinced me of the unexpected findings that not only do seamounts exert an effect on the bacterial community composition up to 200 meters above their summit, but also a more subtle effect on the bacterial community composition of sponges growing at various depths

on the seamount.

The scientific methods are clearly described and appropriate for the work. CTD data were collected at stations both on and off the seamount, so there are appropriate mid-water and near-bottom water controls to assess the influence of the seamount. A sufficient number of sponge samples were collected for each species, which is not a small matter for deep-sea work. Moreover, the authors have put a tremendous amount of work into creating excellent visualizations of the data that clearly show how the results support their conclusions (particularly Figures 3 and 6). It takes real effort to combine such large amounts of information into figures and still have them clearly and cleanly illustrate the narrative points you are making in the discussion.

The discussion is very concise and logically structured. The conclusions are well supported by the results. The references are appropriate in scope, including very recent findings (2018-2019) as well as citing classic papers from multiple decades (centuries). All of the references cited in the text are listed in the bibliography.

I don't often have the pleasure of reviewing a paper that is so articulate and well organized and makes such surprising findings. This manuscript is excellent.

Typographic issues:

Abstract: Line 17: 'extend' should be 'extent' Line 19: 'which extend' should be 'what extent' Line 21: amplicon does not need to be capitalized Lines 38 and 42: remove the comma after 'both' Lines 40-41: You do not define HMA or LMA prior to usage

Introduction: Lines 65-66: 'Rix, De Goeij et al. 2016' should be either Rix et al. 2016a or Rix et al. 2016b Line 71: 'Intimate sponge-microbe associations have been observed. . .' Line 85: '. . .interactivity are still lacking'

Results: Line 242: 'The dominant microbial phylum in S. rosea and L. complicate was Proteobacteria'

Throughout: There is inconsistency of the in-text citations throughout the manuscript.

Some citations italicize 'et al.' while others do not. Please check the journal's preference and then make all of them the same.

References: Line 463: Van Haren is listed under 'H' instead of 'V' Lines 452-457: two De Goeij references are listed under 'G' instead of under 'D' Line 569: Isme should be ISME

---

## Referee Comment (RC2) · Anonymous Referee #2 · 16 Mar 2020

General evaluation

The paper describes pelagic microbial communities at a seamount and in the surrounding deep sea and compares these communities with sponge-associated microbial communities at the same locations. The authors conclude that biogeochemical properties of the water column and hydrodynamic effects induced by the seamount topography may shape these communities and explain differences between seamount summit, flanks and far field stations. These aspects have rarely, if ever, been investigated at seamounts, and the study is an important and interesting contribution to the knowledge of seamount ecology. Although the paper is generally well written, it lacks

some important information regarding methodological aspects, the results are not always presented precisely, and the discussion is superficial in parts, with the results not being fully exploited. For example, the samples were taken in three consecutive years, which could have biased the results considerably, but it is nowhere mentioned which samples were taken in which year and how they differed, and interannual variability and its possible consequences are not discussed. The results of the hydrographic measurements are difficult to see; Fig. 3 is not helpful in this respect. The discussion of relative abundances focuses on only one phylum, but the interesting overall patterns are not considered. There are more issues throughout the text; details are given below. I think that a major revision can improve the paper considerably.

Detailed comments

Abstract

Line 28: I think it should read "at least 200 m", since only two depths were sampled L.40/41: explain abbreviations HMA and LMA

Introduction

L46: although seamounts are widely recognised meanwhile, I would suggest to include a definition here L53: "stimulated primary productivity": insert citation here. This hypothesis has rarely been verified

Methods

L100-108: a figure /map of the seamount location should be included here. Figure 2C could be used as an inset to illustrate the bathymetry. More information is necessary of seamount features: base depth, shape L104: Why this isolated presentation of the near-sea bed temperature? The temperature distribution as derived from the CTD should be presented in the Results, including the allocation of water masses (see also below) L105: I do not quite understand what "minimal values" means in this context L111: See also comment in the general evaluation. How were the samples distributed

across the three years, which subset was taken in which year? This is essential information, since it is well known that temporal variability of water column properties is very high at different scales, and particularly at seamounts with their highly dynamic hydrographic regime. . L113: Again, a map of the sampling locations, including the sponge samples, should be included here, for example based on Fig. 2C. Looking at this figure, all stations were aligned, with some variation, along a W-E axis, with one exception, and I cannot see that any samples were taken along the 74°N-latitude. L117: replace "which" with "and". L117-120: This statistic is strange here. First of all, it is not clear how an ANOVA can be applied to single measurements, but obviously some stations were pooled; the reason and which stations are not provided here (this becomes more clear only later). And even in this case, means and SEs (and hence an ANOVA) make no sense here, because the measurements of depth are no independent replicate measurements of a population, but are just taken at different locations. And, of course, it is trivial that the depth at the summit is shallower than at the flanks and at the base... That's how these regions are defined. Here, just the depth ranges should be indicated. L121: Were these sponges sampled at the same time as the water samples? The number of sponges sampled (i.e. four in each BW1-4 as presented later) indicates that a targeted sampling was done in the subareas defined by the microbial clusters of the near-seabed samples; i.e. probably much later. This information is important for the interpretation. L150: see also comment above. Without knowing the results, it is not clear what is meant here as "sampling region". L186: the purpose of this correlation matrix is not clear. It is not dealt with in the discussion.

Results

L192ff: The extrapolations based on "machine learning" in the contour plots of Fig. 2A and 2B appear very arbitrary, e.g. the N-S extension of MW1 in Fig. 2A, or, even worse, in Fig. 2B, where e.g. BW2 and BW3 extend far into a region which was not covered by samples and features different bathymetric (and most probably also hydrographic) conditions - this is highly unlikely. Even BW1, which was obviously found at

only one station, appears to be present also in in patches in the south and in the north. These extrapolations are confusing and also unnecessary for the interpretation of the results. I suggest to omit the extrapolations and show only the station dots with their respective colours indicating the allocations to the clusters. Line 200: see above; Fig. 2C should be presented in 2.1 L203: replace "overlaying" with "overlying". L202-206: Fig. 3 is too complicated, and the additional results (oceanographic setting) cannot be adequately deduced from the figure. I suggest to provide either simpleT/D-plots, or a 2-dimensional contour plot of temperature with a clear indication of water masses along the main sampling axis. The figure may be useful for interpretations, but then in the discussion section L209: this is not quite logical; the exception from the biodiversity in MW being lower than in BW would be a higher (or equal) biodiversity in MW, but not a difference between BW samples. Why are no data presented here like for the overall richness in BW and MW samples? L212: does this apply only to the summit or to all regions? L213: the difference between this analysis and the one before is not clear. What is "pool" in this respect, and how did these differ? L218: be precise: obviously not samples, but sample regions defined by microbial clustering were compared. L224-234: it would be interesting to see which clusters differed from each other in their biogeochemical properties (e.g. pairwise comparisons). Acc. to Fig. 2B, cluster BW1 consists of only one sample; how was this considered in the ANOVAs? L226: "increased with depth" L223: Here, only the summit stations were compared with respect to their biogeochemical parameters. What about the other locations? L248: Was this correlation with depth statistically tested? How? L255: Interestingly, looking at Fig. 6, Protobacteria had a much higher relative abundance in BW1 than in the other clusters, whereas Gemmatimonadetes had much lower abundance in BW1, but in both phyla differences were not significant. Is there an explanation? In this context, it would be very interesting which clusters differed from each other. For example, Fig. 6 suggests that the differences were mainly between BW1 and the other clusters, which showed only small differences. Could this be tested? L258: see also comment in M&M. This information is not further used, and it is hardly or not at all discernible from Fig. 6.

L264: where is this analysis (correlation between biogeochemical parameters and relative abundances), and how was the statistics done (was this correlation independently tested?)? In Fig. 6, only some relationship between depth and relative abundances is indicated, with differences between depths always corresponding to differences between clusters L266: it is not clear what "significant variability" means in this context, and how this variability was tested L267: which pattern?

Discussion L276: But according to the results (L212ff) community clusters were significantly different between BW and MW samples. This contradiction has to be resolved. L284: Since only one LW depth was sampled, this process could extend far higher than the 200 m, so it should better read "at least". But the process may not have necessarily been restricted to the summit, because due to the much greater distance between LW and BW samples at the other stations, a similar effect may just not have been detected. L291: this applies also to the southern hemisphere! L293: this is far from clear and cannot be deduced from Fig. 3. Apart from the separated clusters at the summit, which may in fact be related to retention and vertical mixing by, e.g., a Taylor column, it is not shown how differences between stations could relate to oscillations of the water column L300: this comparison is hardly applicable here. The Morato et al paper deals with large pelagic predators, and their enhanced biodiversity at seamounts, which is not restricted to the summits, has underlying mechanisms completely different from microbial communities. L307: in which respect do they change? Some information would be helpful (without needing to consult the literature) L310: These are not discernible in Fig. 3. See also comment in the General Evaluation concerning Fig. 3 L312: it is not clear what is meant by "dense ecosystems" L316: include "probably" before "based" - there is no direct evidence L350: "were positively correlated with depth..." No correlation analysis was done between biogeochemical parameters and depth, but discrete ANOVAs for each parameter which revealed differences between cluster regions. These appeared to covary with depth. L353: where does this prediction come from? L354: this ("typically carry more oxygen...") is redundant to the statement before L361: is it really depth (pressure?) that structures the communities,

or depth-associated parameters? L363-370: Why are only Chloroflexi in G. hentscheli discussed? What about the other phyla and sponge species? Particularly with G. hentscheli, there are some interesting patterns which should be discussed in more detail. For example, whereas the relative abundanes of most taxa are very similar in the BG2-BG4 clusters, the abundance of Chloroflexi and Acidobacteria are much lower in BW1, whereas Protobacteria are much higher - is there any explanation? L363: Acc. to Fig. 6, there is a big difference between BW 1 and the other clusters, which are very similar to each other. Has it been tested which clusters differ from each other? L364: I cannot find any results about a positive correlation between these parameters and the clusters L367-369: this is not quite clear here. Usually, the oxygen demand is enhanced by the (microbial) degradation of OM, which on the other hand sets free nutrients such as $NO_3-$ and may enhance denitrification processes. It would be interesting in this respect to learn something about the metabolic pathways of Chloroflexi, e.g. whether they are involved in denitrification, which could explain a positive correlation with $NO_3-$.

Conclusions

L378:"high resolution sampling" is rather meaningless - distributing sampling over three years is rather not high resolution, and whether the spatial resolution is high is also questionable. I suggest to omit this; it is not necessary. 380: "has a detectable but variable influence. . ." I would be careful with this statement. There appeared to be some interrelation (a statistical correlation was not shown), but it could not be convincingly shown that a causal relationship with those parameters was highly likely, or which of the three was probably the key parameter. A possible mechanistic explanation would be interesting, for example with respect to metabolic functioning of the microbial phyla. What about interannual variability - the paper does not provide any information that would rule out a possible effect of the sampling dates.

Figures

Fig. 2: see comments in Results. Fig. 2C belongs into the Methods section

Fig. 3: This Figure should be placed into the Discussion and help interpreting the results. It is not suitable for the presentation of results, because, for example, the temperature profiles and water mass distribution are not readily identifiable in the 3D setting

Fig. 4: y-axis labelling is missing. Degrees of freedom of the ANOVAs should be included.

Fig. 6: the correlation matrix should be omitted - it is not used and is hardly (A and B) or not all discernible (C and D). Panel C: Geodia in italics.

---

## Author Comment (AC1) · 29 Mar 2020

**Response to Reviewer Comments on Manuscript bg-2020-15:**

**On giant shoulders: How a seamount affects the microbial community composition of seawater and sponges**

Kathrin Busch, Ulrike Hanz, Furu Mienis, Benjamin Müller, Andre Franke, Emyr Martyn Roberts, Hans Tore Rapp, Ute Hentschel

*We thank the reviewers for the overall positive evaluation of our manuscript and for the time and effort taken. We have made considerable effort to address the points raised and also included three additional figures (Supplementary Material S1, Supplementary Material S2A, Supplementary Material S3), elaborated existing figures (Fig.3, Fig.4, Fig.6) and conducted further statistical analyses (i.e. TukeyHSD Posthoc tests, Spearman´s rank correlation). Below please find our responses that are listed in the order raised. We show the referees' comments in black text, while our responses are formatted in red. The new line numbers refer to the revised manuscript (marked-up version). Thank you for your consideration.*

**Reviewer #1 (Comments for the authors):**

**Overall comments:** This manuscript brings to bear physical oceanographic, biogeochemical, and microbiological data on the question of whether seamounts impact the microbial community composition of the water column and benthic organisms like sponges. I was skeptical of water column impacts, imagining that a given water mass would have a microbial signature irrespective of the seamount. However, the data in this paper convinced me of the unexpected findings that not only do seamounts exert an effect on the bacterial community composition up to 200 meters above their summit, but also a more subtle effect on the bacterial community composition of sponges growing at various depths on the seamount. The scientific methods are clearly described and appropriate for the work. CTD data were collected at stations both on and off the seamount, so there are appropriate mid-water and near-bottom water controls to assess the influence of the seamount. A sufficient number of sponge samples were collected for each species, which is not a small matter for deep-sea work. Moreover, the authors have put a tremendous amount of work into creating excellent visualizations of the data that clearly show how the results support their conclusions (particularly Figures 3 and 6). It takes real effort to combine such large amounts of information into figures and still have them clearly and cleanly illustrate the narrative points you are making in the discussion. The discussion is very concise and logically structured. The conclusions are well sup-ported by the results. The references are appropriate in scope, including very recent findings (2018-2019) as well as citing classic papers from multiple decades (centuries). All of the references cited in the text are listed in the bibliography. I don't often have the pleasure of reviewing a paper that is so articulate and well organized and makes such surprising findings. This manuscript is excellent.
Typographic issues:

**Abstract:**

Line 17: 'extend' should be 'extent'
DONE

Line 19: 'which extend' should be 'what extent'
DONE

Line 21: amplicon does not need to be capitalized
DONE

40

Lines 38 and 42: remove the comma after 'both'
DONE

45 Lines 40-41: You do not define HMA or LMA prior to usage
The explanation of HMA or LMA as well as the taxonomic status has been removed here as it is explained later (LL79-81).

**Introduction:**

Lines 65-66: 'Rix, De Goeij et al. 2016' should be either Rix et al.2016a or Rix et al. 2016b
DONE

50

Line 71: 'Intimate sponge-microbe associations have been observed. . .'
DONE

Line 85: '. . .interactivity are still lacking'
55 DONE

**Results:**

Line 242: 'The dominant microbial phylum in S. rosea and L. complicate was Proteobacteria'
DONE

60 Throughout: There is inconsistency of the in-text citations throughout the manuscript. Some citations italicize 'et al.' while others do not. Please check the journal's preference and then make all of them the same.
DONE

**References:**

Line 463: Van Haren is listed under 'H' instead of 'V'
65 DONE

Lines 452-457: two De Goeij references are listed under 'G' instead of under 'D'
DONE

70 Line 569: Isme should be ISME
DONE

75

[Figure]

**(Revised) Figure 3: Conceptual overview and vertical 3D section showing spatial distribution of microbial clusters and oceanographic patterns on the Schulz Bank seamount.** Extrapolated seawater microbial clusters are indicated by colored polygons: mid-water clusters are marked in orange (MW1) and yellow (MW2), while near-bed water clusters are marked in red (BW1), dark blue (BW2), light blue (BW3), and purple (BW4). The degree of cluster similarity can be deduced from the dendrogram in the left corner of the plot. Whole water column CTD profiles are indicated, showing the measured temperature values from surface to bottom at the respective sampling locations. Sponge sampling locations visible on this side of the seamount are indicated by white balls. Vertical positions of major watermasses: Norwegian Atlantic Water (NwAtW), Norwegian Arctic Intermediate Water (NwArIW), and Norwegian Deep Water (NwDW) are indicated. To give a broad orientation in space, a north arrow is depicted, as well as the major geologic features (Lofoten Basin and Greenland Basin). For Schulz Bank, water flows, such as a potential Taylor column circulating around the seamount, mixing between summit and pelagic realm, as well as tidally-driven internal motions (black horizontal line with bidirectional arrows) are indicated by dashed arrows and lines.

[Figure]

105

(Revised) Figure 4: Concentrations and measurements of significant (ANOVA, α=0.05) biogeochemical parameters for near-bed water samples, across the determined near-bed water clusters. p-values as well as degrees of freedom (df) for these parameters are written into the respective graphs. Colouring is the same as chosen for Fig. 2 and 3.

[Figure]

110 **(Revised) Figure 6: Co-occurrence network and differential abundance of microbial phyla across the four determined near-bed water clusters. Sub-plots A-C) show sponge data, with plot A) showing average *Schaudinnia rosea* data, B) presenting average *Lissodendoryx complicata* data and C) illustrating *Geodia hentscheli* data. Sub-plot D) contains seawater data. Near-bed water clusters are represented by differently colored rings. Each ring contains a list with microbial phyla which are alphabetically sorted. Average relative abundances of each of the respective phyla for the samples within a given cluster are indicated by bubble sizes.**

115 **Those microbial phyla which are statistically significantly enriched or depleted across the four clusters (LEfSe analysis), are marked with an asterisk inside the inner most ring. Only for those taxa where the difference is significant, correlation strength (indicated by size of connecting lines) and direction (represented by color of connecting lines: white = negative correlation, dark grey = positive correlation) with all other significant taxa are plotted. For all microbial phyla relation with depth is indicated in the outer ring of each plot by + (meaning significant positive relation) or - (meaning significant negative relation).**

[Figure]

**(Newly added) Supplementary Figure S1: Overview map showing the location of the Schulz Bank seamount between the Greenland and Norwegian Seas.**

[Figure]

(Newly added) Supplementary Figure S2: A) Overall richness of pooled mid-water microbial communities and pooled near-bed water microbial communities. Faith Phylogenetic Diversity (Faith's PD) is plotted as alpha-diversity measure. B) Richness of seawater microbial communities for the mid-water and near-bed water samples, across the determined clusters. Mid-water clusters are coloured in orange (MW1) and yellow (MW2), while near-bed water clusters are marked in red (BW1), dark blue (BW2), light blue (BW3), and purple (BW4).

130

[Figure]

[Figure]

**(Newly added) Supplementary Figure S3: Spearman's rank correlations calculated between depth and the three biogeochemical parameters which differed significantly (ANOVA, α=0.05) across the determined near-bed water clusters.**

---

## Author Comment (AC2) · 29 Mar 2020

**Response to Reviewer Comments on Manuscript bg-2020-15:**

**On giant shoulders: How a seamount affects the microbial community composition of seawater and sponges**

Kathrin Busch, Ulrike Hanz, Furu Mienis, Benjamin Müller, Andre Franke, Emyr Martyn Roberts, Hans
Tore Rapp, Ute Hentschel

*We thank the reviewers for the overall positive evaluation of our manuscript and for the time and effort taken. We have made considerable effort to address the points raised and also included three additional figures (Supplementary Material S1, Supplementary Material S2A, Supplementary Material S3), elaborated existing figures (Fig.3, Fig.4, Fig.6) and conducted further statistical analyses (i.e. TukeyHSD Posthoc tests, Spearman´s rank correlation). Below please find our responses that are listed in the order raised. We show the referees' comments in black text, while our responses are formatted in red. The new line numbers refer to the revised manuscript (marked-up version). Thank you for your consideration.*

**Reviewer #2 (Comments for the authors):**

**General evaluation:** The paper describes pelagic microbial communities at a seamount and in the surrounding deep sea and compares these communities with sponge-associated microbial communities at the same locations. The authors conclude that biogeochemical properties of the water column and hydrodynamic effects induced by the seamount topography may shape these communities and explain differences between seamount summit, flanks and far field stations. These aspects have rarely, if ever, been investigated at seamounts, and the study is an important and interesting contribution to the knowledge of seamount ecology. Although the paper is generally well written, it lacks some important information regarding methodological aspects, the results are not always presented precisely, and the discussion is superficial in parts, with the results not being fully exploited. For example, the samples were taken in three consecutive years, which could have biased the results considerably, but it is nowhere mentioned which samples were taken in which year and how they differed, and interannual variability and its possible consequences are not discussed. The results of the hydrographic measurements are difficult to see; Fig. 3 is not helpful in this respect. The discussion of relative abundances focuses on only one phylum, but the interesting overall patterns are not considered. There are more issues throughout the text; details are given below. I think that a major revision can improve the paper considerably.
Detailed comments:

**Abstract:**

Line 28: I think it should read "at least 200 m", since only two depths were sampled
DONE

L.40/41: explain abbreviations HMA and LMA
The explanation of HMA and LMA has been removed from the abstract as it is given in the Introduction (LL79-81).

**Introduction:**

L46: although seamounts are widely recognised meanwhile, I would suggest to include a definition here
A definition was added (LL48-51).

L53: "stimulated primary productivity": insert citation here. This hypothesis has rarely been verified
A reference was added (L59).

**Methods:**

L100-108: a figure /map of the seamount location should be included here. Figure2C could be used as an inset to illustrate the bathymetry.
A map of the seamount location has been added (Supplementary Material S1).
Concerning Figure2C please see comment below where we explain that we prefer to keep Fig.2C together with Figs 2A and 2B.

More information is necessary of seamount features: base depth, shape
DONE (LL114-116).

L104: Why this isolated presentation of the near-sea bed temperature? The temperature distribution as derived from the CTD should be presented in the Results, including the allocation of water masses (see also below).
The near-sea bed temperature and the allocation of water masses are provided by Roberts et al (2018) and are not our own results. We have moved the sentence before the reference Roberts et al (2018) so that this becomes clearer (LL111-112).

L105: I do not quite understand what "minimal values" means in this context
This sentence has been clarified (L113).

L111: See also comment in the general evaluation. How were the samples distributed across the three years, which subset was taken in which year? This is essential in-formation, since it is well known that temporal variability of water column properties is very high at different scales, and particularly at seamounts with their highly dynamic hydrographic regime.
All of the requested information along with an interactive map of the seamount and individual references on CTD and sponge samplings has been made publicly available on PANGAEA at the time of manuscript submission. Pls follow the link (L 430) to find this data depository.

L113: Again, a map of the sampling locations, including the sponge samples, should be included here, for example based on Fig. 2C.
See PANGAEA for the requested information. We think that this interactive visualization (PANGAEA) is more informative than a classical graph, as individual sampling points of seawater and sponges lay in very close vicinity to each other and overlap sometimes.

Looking at this figure, all stations were aligned, with some variation, along a W-E axis, with one exception, and I cannot see that any samples were taken along the 74∘N-latitude.
The 74°N value has been corrected to 73.8°N (L124).

L117: replace "which" with "and".
DONE (L127).

L117-120: This statistic is strange here. First of all, it is not clear how an ANOVA can be applied to single measurements, but obviously some stations were pooled; the reason and which stations are not provided here (this becomes more clear only later). And even in this case, means and SEs (and hence an ANOVA) make no sense here, because the measurements of depth are no independent replicate measurements of a population, but are just taken at different locations. And, of course, it is trivial that the depth at the summit is shallower than at the flanks and at the base. . .That's how these regions are defined. Here, just the depth ranges should be indicated.

Thank you for raising this issue, this is indeed confusing. Once the MW and BW microbiome clusters were identified based on similarity of microbial communities irrespective of any metadata, we queried where they would fall into horizontal and vertical space. We found – based on ANOVA analyses of the individual depth data points underlying each cluster – that the clusters are indeed significantly different in depth. In fact, we consider depth as an independent replicate measurement at Schulz Bank. The sentences have therefore been moved from the methods into the results section (LL216-220).

L121: Were these sponges sampled at the same time as the water samples? The number of sponges sampled (i.e. four in each BW1-4 as presented later) indicates that a targeted sampling was done in the subareas defined by the microbial clusters of the near-seabed samples; i.e. probably much later. This information is important for the interpretation.

Sponges were sampled at the same time as the water samples. We had collected more sponges, of which a random subset was chosen so that the sampling design was balanced for statistical purposes. This is stated more clearly now (LL132-133).

L150: see also comment above. Without knowing the results, it is not clear what is meant here as "sampling region".

This is correct. The sentence was moved into the results after the clusters have been identified (LL261-262).

L186: the purpose of this correlation matrix is not clear. It is not dealt with in the discussion.

Please see our argumentation below.

**Results:**

L192ff: The extrapolations based on "machine learning" in the contour plots of Fig.2A and 2B appear very arbitrary, e.g. the N-S extension of MW1 in Fig. 2A, or, even worse, in Fig. 2B, where e.g. BW2 and BW3 extend far into a region which was not covered by samples and features different bathymetric (and most probably also hydrographic) conditions - this is highly unlikely. Even BW1, which was obviously found at only one station, appears to be present also in in patches in the south and in the north. These extrapolations are confusing and also unnecessary for the interpretation of the results. I suggest to omit the extrapolations and show only the station dots with their respective colours indicating the allocations to the clusters.

We do not agree to remove this important figure which is key to our findings. While we have accrued a considerable amount of data on this remote location, machine learning is a very useful tool that allows us to expand our predictions into regions that could not be/may never be obtained by hands-on sampling. We are aware that these are "bioinformatic predictions" and have indicated that clearly in the legend. To make this predictive value even clearer, we have added the following sentence: "The further away predicted areas are from actual sample points, the higher is the associated uncertainty of these predictions."

Line 200: see above; Fig.2C should be presented in 2.1

We prefer to keep Fig. 2C here because it serves as an important reference point for Figs 2A and 2B.

L203: replace "overlaying" with "overlying".

DONE (L224).

L202-206: Fig. 3 is too complicated, and the additional results (oceanographic setting) cannot be adequately deduced from the figure. I suggest to provide either simple T/D-plots, or a 2-dimensional contour plot of temperature with a clear indication of water masses along the main sampling axis. The figure may be useful for interpretations, but then in the discussion section

We feel strongly that this conceptual overview should remain in the manuscript. It presents a 3D visualisation of microbiome clusters as derived by machine learning in the context of real oceanographic data that have either been collected (CTDs) or

that have already been published (water masses) using the exact same T/S data of the 2016 cruise (Roberts et al. 2018). Simplification is not possible without the loss of data. We propose that it should remain here so that the connection to the 2D visualization (Fig. 2) using the same color scheme is maintained.

130

L209: this is not quite logical; the exception from the biodiversity in MW being lower than in BW would be a higher (or equal) biodiversity in MW, but not a difference between BW samples. Why are no data presented here like for the overall richness in BW and MW samples?

135    Correct. The sentence has been changed (L228-231). A plot showing the overall richness in BW and MW samples has been added (Supplementary Material S2A).

L212: does this apply only to the summit or to all regions? L213: the difference between this analysis and the one before is not clear. What is "pool" in this respect, and how did these differ?

140    The sentence (LL232-236) has been modified. The seawater microbial community clusters within the two groups (MW, BW) were significantly different from each other in terms of their microbial community composition. Moreover, the pool of mid-water samples (MW1 + MW2) was significantly different from the pool of near-bed water samples (BW1 - BW4). This has been stated more clearly.

145    L218: be precise: obviously not samples, but sample regions defined by microbial clustering were compared.
Correct. The term mid-water samples have been replaced with mid-water clusters (L241).

L224-234: it would be interesting to see which clusters differed from each other in their biogeochemical properties (e.g. pairwise comparisons). Acc. to Fig. 2B, cluster BW1 consists of only one sample; how was this considered in the ANOVAs?

150    We would like to clarify that cluster BW1 consists of three microbial samples that are overlaying in Fig. 2B. However, the reviewer is right that for BW1 only one biogeochemical sample is included in the ANOVA analyses. The sample numbers in the other BWs are consistently higher. As the ANOVA design was thus unbalanced, we calculated Type III sums of squares to account for this aspect while performing ANOVAs. We added an according remark for clarification into the manuscript (LL158-161).

155    Standard Posthoc tests (TukeyHSD, Bonferroni, etc) are generally sensitive against unequal sample sizes. However, following the reviewer´s suggestion, we run TukeyHSD tests (based on linear model fits) for those parameters which turned out to be significant in ANOVA analyses (i.e. depth, oxygen, nitrate, silicate). When doing so, we only observed one significant difference of BW1 in comparison to the other BWs (for nitrate). The most significant difference was between BW3-BW4, which are also the clusters with the highest samples numbers. We think that these Posthoc test results are strongly biased by

160    technical issues due to unequal sample sizes. We conclude that those tests are not helping in improving the manuscript content-wise. Trends to answer this question for the interested reader can however be deduced from the boxplots (Fig.4) included in the manuscript.

General remark: In respect to the limited number of biogeochemical samples for BW1, we agree that the design could have been improved with biogeochemical measurements at additional sites across Schulz Bank. However, as we are – to the best of

165    our knowledge – the first to characterize the microbial seawater community of Schulz Bank (and also among the first seamount pelagic microbial studies globally) we could not know already during sampling if our (already higher) spatial resolution at the summit would be sufficient. As reviewer#1 states, we have compiled a considerable number of samples for deep-sea work. Due to expensive ship time also more sampling within one year would not have been possible, but we see and acknowledge (see Pangaea) the respective limitations that the reviewer brings up.

170

L226: "increased with depth"
DONE (L250).

L223: Here, only the summit stations were compared with respect to their biogeochemical parameters. What about the other
175    locations?
We only had two clusters in the mid-water (LL241-246). The near-bed water clusters were compared in the following paragraph (LL247-257) and further details are given in Supplementary Table S2.

L248: Was this correlation with depth statistically tested? How?

We determined microbial taxa that differed significantly across the near-bed water clusters by LEfSe analyses. The identified microbial clusters of the near-bed water are obviously categorical explanatory variables (and not continuous variables). Therefore we did not perform a correlation between the determined microbial clusters and any microbial phylum. Depth turned out to be among those environmental parameters that differed significantly across the near-bed water clusters. In Fig.6 we use depth as a proxy for the microbial clusters and the other three significant biogeochemical parameters to minimize complexity for the reader and give an ecological context. We have included correlation analyses in the revised manuscript, which clarify that there is indeed a significant correlation between depth and those biogeochemical parameters.

Concerning the "correlation with depth" (e.g. L275) we are referring to the following: To assess "correlation with depth", we plotted for all samples of every sponge species the relative abundances of each microbial phylum across all near-bed water clusters (boxplots with near-bed cluster on x-axis and relative abundance of the phylum on y-axis). By visual inspection we determined for those taxa turning out as significant in the LEfSe analysis, if they follow the same profile as depth (boxplot Fig.4).

We changed the term "correlation" to the term "relation" throughout the manuscript, where it was used in this context. Correlation now only refers to a statistically tested correlation sensu stricto (i.e. for example Spearman correlation).

L255: Interestingly, looking at Fig. 6, Protobacteria had a much higher relative abundance in BW1 than in the other clusters, whereas Gemmatimonadetes had much lower abundance in BW1, but in both phyla differences were not significant. Is there an explanation?

We are reporting here on those phyla that were statistically significantly different between the BW clusters. One explanation is that statistical significance will depend not solely on the abundance in one particular BW cluster but across all four BW clusters.

In this context, it would be very interesting which clusters differed from each other. For example, Fig. 6 suggests that the differences were mainly between BW1 and the other clusters, which showed only small differences. Could this be tested?

Please see our comment above about pairwise comparisons. In brief, we are limited in the number of biogeochemical samples particularly for BW1 as the reviewer has realized. We provide an argumentation above why we think our non-classical study design is leading to valuable, novel insights and prefer to put the focus of this study on overall trends across clusters, instead of pairwise considerations between clusters.

L258: see also comment in M&M. This information is not further used, and it is hardly or not at all discernible from Fig. 6.

We have added sentences evaluating on this result in the Discussion (LL383-386). Thanks for bringing this to our attention, as we think that this result is indeed very interesting. Based on this result we suspect that primary responders to environmental parameters have cascading effects on microbial lineages that are not directly affected by water biogeochemistry.

Concerning the suggested omission of the correlation matrix, we are the first to compute co-occurrence networks for deep-sea sponge microbiomes. It illustrates that different numbers and types of microbial taxa vary across the near-bed clusters in every sponge species and in seawater. This information is valuable for more mechanistic follow-up analyses.

L264: where is this analysis (correlation between biogeochemical parameters and relative abundances), and how was the statistics done (was this correlation independently tested?)? In Fig. 6, only some relationship between depth and relative abundances is indicated, with differences between depths always corresponding to differences between clusters

Please consider our detailed comment above. Depth is here taken as a proxy, because the detailed statistical analysis of relative abundances within each phylum in each sponge species against each biogeochemical parameter would be beyond the scope of this study. Here, we report on the general observation, which is novel and exciting as both reviewers acknowledge.

L266: it is not clear what "significant variability" means in this context, and how this variability was tested. L267: which pattern?

These two points have been clarified: The microbial taxa showing a significant difference in relative abundance between the near-bed water clusters (as determined by LEfSe) were different between sponges and seawater also between sponge species.

Further, significant differences in relative abundances were observed for both abundant and less abundant sponge symbiont lineages (LL293-296).

**Discussion:**

L276: But according to the results (L212ff) community clusters were significantly different between BW and MW samples. This contradiction has to be resolved.
With this statement, we are referring to the few black spots (representing BW1) that cluster with MW samples (Fig. 1). Even though these BW samples are more similar to the MW samples, they don't have to be identical. The wording has been revised accordingly (LL309-310).

L284: Since only one LW depth was sampled, this process could extend far higher than the 200 m, so it should better read "at least". But the process may not have necessarily been restricted to the summit, because due to the much greater distance between LW and BW samples at the other stations, a similar effect may just not have been detected.
The wording has been changed to "at least" (L317). While we agree that similar effects may apply to other depths, we prefer here to discuss the presented data.

L291: this applies also to the southern hemisphere!
To our knowledge, anti-cyclonic circulation patterns (as indicated by the counter-clock wise arrow in Fig. 3) are specific for the Northern hemisphere. We prefer to leave the sentence unchanged.

L293: this is far from clear and cannot be deduced from Fig. 3. Apart from the separated clusters at the summit, which may in fact be related to retention and vertical mixing by, e.g., a Taylor column, it is not shown how differences between stations could relate to oscillations of the water column.
The oscillations relate to tidal-induced variations in the water column structure, hence do not reflect differences between stations but rather similarities between mid-water and near-bed water samples at the summit. Please consider our interpretation in LL315-319. We have included a clustering dendrogram into the legend of Fig. 3 to clarify similarities and dissimilarities of microbial communities between the water clusters. Further we have added arrows for clarification of tidal-induced water movements.

L300: this comparison is hardly applicable here. The Morato et al paper deals with large pelagic predators, and their enhanced biodiversity at seamounts, which is not restricted to the summits, has underlying mechanisms completely different from microbial communities.
The sentence has been reworded (LL331-333).

L307: in which respect do they change? Some information would be helpful (without needing to consult the literature)
The sentence has been reworded (LL340-341).

L310: These are not discernible in Fig. 3. See also comment in the General Evaluation concerning Fig.3
Clarification has been added (LL343-345).

L312: it is not clear what is meant by "dense ecosystems"
The sentence has been reworded (LL346-349).

L316: include "probably" before "based" - there is no direct evidence
DONE (L351).

L350: "were positively correlated with depth. . ." No correlation analysis was done between biogeochemical parameters and depth, but discrete ANOVAs for each parameter which revealed differences between cluster regions. These appeared to covary with depth.

Thanks for this thoughtful suggestion. We calculated Spearman´s rank correlation coefficients between depth and those biogeochemical parameters which turned out to differ significantly across the determined near-bed water clusters in the ANOVA analyses (LL159-161; LL253-255; S3).

L353: where does this prediction come from?
"Prediction" has been replaced with "expectation" (L392).

L354: this ("typically carry more oxygen. . .") is redundant to the statement before
The first statement refers to the process, the latter to concentration; this is an accurate sentence in our opinion.

L361: is it really depth (pressure?) that structures the communities, or depth-associated parameters?
We are here citing Indraningrat et al (2019) and prefer to be consistent with the authors' wording.

L363-370: Why are only Chloroflexi in G. hentscheli discussed? What about the other phyla and sponge species? Particularly with G.hentscheli, there are some interesting patterns which should be discussed in more detail. For example, whereas the relative abundances of most taxa are very similar in the BG2-BG4 clusters, the abundance of Chloroflexi and Acidobacteria are much lower in BW1, whereas Protobacteria are much higher - is there any explanation?
We have chosen this phylum as an example, because Chloroflexi are abundant and representative sponge symbionts, because the relative abundances are co-varying with depth, and because we show data for all BW clusters in *G. hentscheli*. We do not see the value to discuss this pattern with all clades because this would enormously inflate the discussion. An explanatory statement along these lines has been added (LL402-406).

L363: Acc.to Fig. 6, there is a big difference between BW 1 and the other clusters, which are very similar to each other. Has it been tested which clusters differ from each other?
The aim of our study is to explore changes in the sponge microbiome across the different near-bed water clusters. As explained above, depth serves as a proxy for the selected biogeochemical parameters that correlate with depth. We did not aim to relate changes in sponge microbiomes with each cluster separately. Please consider also our comments above.

L364: I cannot find any results about a positive correlation between these parameters and the clusters
Please see our comments above.

L367-369: this is not quite clear here. Usually, the oxygen demand is enhanced by the (microbial) degradation of OM, which on the other hand sets free nutrients such as NO3- and may enhance denitrification processes. It would be interesting in this respect to learn something about the metabolic pathways of Chloroflexi, e.g. whether they are involved in denitrification, which could explain a positive correlation with NO3-.
We are not going into functions of individual phyla because the biological interactions between the sponge host and its microbiome, or between the microbes themselves might have masking effects. Pls see our statement (L417-418).

**Conclusions:**

L378:"high resolution sampling" is rather meaningless - distributing sampling over three years is rather not high resolution, and whether the spatial resolution is high is also questionable. I suggest to omit this; it is not necessary.
The sentence has been reworded (LL420-421).

L380: "has a detectable but variable influence. . ." I would be careful with this statement. There appeared to be some
interrelation (a statistical correlation was not shown), but it could not be convincingly shown that a causal relationship with
those parameters was highly likely, or which of the three was probably the key parameter. A possible mechanistic explanation
would be interesting, for example with respect to metabolic functioning of the microbial phyla. What about interannual
variability - the paper does not provide any information that would rule out a possible effect of the sampling dates.
"Interannual variability": Please note that during the sampling process, we always sampled seawater references with a time
lag of maximum a few hours to according sponge sampling. More details on the sampling dates have been made publicly
available in the Pangaea database before submission.
"Variable influence": We are referring to the observation that we observed no uniform patterns for all three analysed sponge
species. Whether these patterns may be individual-specific or species-specific and which environmental parameter is the key
parameter, exceeds the scope of this study.
We are reporting here correlation but not causation or mechanistic relations/functions between sponge microbiomes and
seawater biogeochemistry. Without deep metagenomic data on functional gene inventories this is not the goal of the present
study.

**Figures:**

Fig. 2: see comments in Results. Fig. 2C belongs into the Methods section
Please see comments above.

Fig. 3: This Figure should be placed into the Discussion and help interpreting the results. It is not suitable for the presentation
of results, because, for example, the temperature profiles and water mass distribution are not readily identifiable in the 3D
setting
The microbiome clusters presented in Fig. 3 come from our own analyses, the flow patterns and oscillations of the water
column (dashed lines) were taken from Roberts et al (2018) who already provided the more detailed data and 2D-visualisations
which the reviewer requests (above). The novelty in our study is the integration of this physical data with the newly generated
microbial data. We have added a clustering dendrogram into Fig.3, which is an overarching result of our study. We prefer to
leave this figure with the results.

Fig. 4: y-axis labelling is missing. Degrees of freedom of the ANOVAs should be included.
Thanks! We added y-axis labels, the degrees of freedom of the ANOVAs (into Fig.4), and adjusted the figure legend
accordingly.

Fig. 6: the correlation matrix should be omitted - it is not used and is hardly (A and B) or not all discernible (C and D).
See comments above.

Panel C: Geodia in italics.
DONE.

[Figure]

360 **(Revised) Figure 3: Conceptual overview and vertical 3D section showing spatial distribution of microbial clusters and oceanographic patterns on the Schulz Bank seamount. Extrapolated seawater microbial clusters are indicated by colored polygons: mid-water clusters are marked in orange (MW1) and yellow (MW2), while near-bed water clusters are marked in red (BW1), dark blue (BW2), light blue (BW3), and purple (BW4). The degree of cluster similarity can be deduced from the dendrogram in the left corner of the plot. Whole water column CTD profiles are indicated, showing the measured temperature values from surface to**

365 **bottom at the respective sampling locations. Sponge sampling locations visible on this side of the seamount are indicated by white balls. Vertical positions of major watermasses: Norwegian Atlantic Water (NwAtW), Norwegian Arctic Intermediate Water (NwArIW), and Norwegian Deep Water (NwDW) are indicated. To give a broad orientation in space, a north arrow is depicted, as well as the major geologic features (Lofoten Basin and Greenland Basin). For Schulz Bank, water flows, such as a potential Taylor column circulating around the seamount, mixing between summit and pelagic realm, as well as tidally-driven internal motions (black**

370 **horizontal line with bidirectional arrows) are indicated by dashed arrows and lines.**

[Figure]

(Revised) Figure 4: Concentrations and measurements of significant (ANOVA, α=0.05) biogeochemical parameters for near-bed water samples, across the determined near-bed water clusters. p-values as well as degrees of freedom (df) for these parameters are written into the respective graphs. Colouring is the same as chosen for Fig. 2 and 3.

390

(Revised) Figure 6: Co-occurrence network and differential abundance of microbial phyla across the four determined near-bed water clusters. Sub-plots A-C) show sponge data, with plot A) showing average *Schaudinnia rosea* data, B) presenting average *Lissodendoryx complicata* data and C) illustrating *Geodia hentscheli* data. Sub-plot D) contains seawater data. Near-bed water clusters are represented by differently colored rings. Each ring contains a list with microbial phyla which are alphabetically sorted. Average relative abundances of each of the respective phyla for the samples within a given cluster are indicated by bubble sizes. Those microbial phyla which are statistically significantly enriched or depleted across the four clusters (LEfSe analysis), are marked with an asterisk inside the inner most ring. Only for those taxa where the difference is significant, correlation strength (indicated by size of connecting lines) and direction (represented by color of connecting lines: white = negative correlation, dark grey = positive correlation) with all other significant taxa are plotted. For all microbial phyla relation with depth is indicated in the outer ring of each plot by + (meaning significant positive relation) or - (meaning significant negative relation).

[Figure]

**(Newly added) Supplementary Figure S1: Overview map showing the location of the Schulz Bank seamount between the Greenland and Norwegian Seas.**

405

(Newly added) Supplementary Figure S2: A) Overall richness of pooled mid-water microbial communities and pooled near-bed water microbial communities. Faith Phylogenetic Diversity (Faith's PD) is plotted as alpha-diversity measure. B) Richness of seawater microbial communities for the mid-water and near-bed water samples, across the determined clusters. Mid-water clusters are coloured in orange (MW1) and yellow (MW2), while near-bed water clusters are marked in red (BW1), dark blue (BW2), light blue (BW3), and purple (BW4).

410

415

(Newly added) Supplementary Figure S3: Spearman's rank correlations calculated between depth and the three biogeochemical parameters which differed significantly (ANOVA, α=0.05) across the determined near-bed water clusters.

---

## Author Response (AR2)

*Kiel, 29st of April 2020*

*Dear Tina Treude (editor),*

*please find below our revised manuscript (marked-up version).*

*Right after this cover letter we have added our point-by-point response to the reviews, including a listing of all relevant changes made in the manuscript.*

*In respect to this manuscript, we also have to inform you that my co-author and mentor Hans Tore Rapp has died unexpectedly during the review process (on the 7th of March 2020).*

*He had expressed his consent with the original manuscript version we submitted and we have added a sign next to his name to indicate that he is deceased.*

*We hope that this is according to the formal requirements of Biogeosciences. If we need to do anything else in this respect, please let us know so that we can take care of it.*

*Kind regards,*

*Kathrin Busch on behalf of the authors*

**2ⁿᵈ Response to Editor and Reviewer Comments on Manuscript bg-2020-15:**

**On giant shoulders: How a seamount affects the microbial community composition of seawater and sponges**

Kathrin Busch, Ulrike Hanz, Furu Mienis, Benjamin Müller, Andre Franke, Emyr Martyn Roberts, Hans Tore Rapp, Ute Hentschel

*To the editor and reviewer # 2,*

*We have followed your advice and revised again our submitted manuscript. In particular we have modified existing figures (Fig. 1, Fig. 4) and created a new figure (Supplementary Material S1). With the latter we aimed to facilitate a trace-back of potential temporal variations (which we argue are only minor in our study) and to provide a presentation of basic sampling information without the need to go to the PANGAEA repository. In addition we have modified the manuscript text following the reviewers suggestions. Our detailed point-to-point answers can be found below. We show the referee´s and editor´s comments in black text, while our responses are formatted in red. The line numbers and figure names in our responses refer to the new (marked-up) manuscript version.*

*With kind regards,*

*Kathrin Busch on behalf of the authors*

**Tina Treude, Editor (Comments to the author):**

**Associate Editor Decision: Reconsider after major revisions** (22 Apr 2020)

Dear Kathrin and Co-Workers, after the review of the revised manuscript, the reviewer has still some comments and requests that warrant a second major revision. However, my understanding is that the required modifications are mostly of technical nature and you should be able to implement them easily. The reviewer is generally very supportive of the study and has a high appreciation of the dataset. I advice you to take the suggestions seriously.

I have a few comments myself in response to the reviewer's suggestion:

- I agree that Fig S1 (map) should be included in the main manuscript. I am a little irritated, however, that instead of a single red dot marking the location of the study site there appear to be an accumulation of dots. Do these mark different locations? If so, please add a zoomed-in insert map to the main map that provides enough resolution to tell the dots apart. If not, then please reduce to just one dot.

Done. We have indicated the location of the seamount's summit in the map and moved Supplementary Material S1 (now Fig. 1) into the main manuscript.
(For clarification: The multiple points indicated in the previous version of the map were those sampling sites along Schulz Bank which are presented more closely in Fig. 3, just with very large circle sizes).

- The reviewer finds that referencing the interactive map provided on PANGAEA is not sufficient and requests to incorporate more details into the actual manuscript. I sympathize with this request. I advice you to extract the minimum amount of information required from PANGAEA to understand the study and provide it in the manuscript. This does not have to be the entire map, just as much information that is required to follow and understand your study. Basically, someone should not be required to go to PANGAEA to follow your study. Data repositories along publications are mainly meant to provide details not required for your interpretations and to allow others to make use of the data in the future.

Done. We have created an additional figure (Supplementary Material S1A-B) accounting for this criticized aspect. Supplementary Material S1A shows which seawater samples were sampled in which year to clarify on the temporal aspect. Supplementary Material S1B depicts locations of sponge sampling in relation to seawater sampling positions.

- The reviewer suggest to move Figure 3 to the discussion, because no new (unpublished) data are presented. If that is the case, I agree with with this suggestion. If, however, new data were incorporated, I would be OK with leaving it in the results but please provide more details of the combination of new and existing datasets that went into the plot.

Done. Figure 3 (now named Fig. 7) has been moved to the discussion.

Let me know in case you have any questions.

Please provide a point-by-point response and a track-changes-version of your manuscript when you submit the revised manuscript.

All the best and stay safe!

Tina

**Reviewer # 2 (Comments for the authors):**

**Evaluation of Busch et al: On giant shoulders: How a seamount affects the microbial community composition of seawater and sponges (bg-2020-15-manuscript-version3).**

The authors improved the manuscript considerably, and the new figures are a welcome addition that make the text better comprehensible. But still, there are a couple of issues left, and the authors' rebuttal to the original comments are not always convincing. Particularly, missing information regarding methodological aspects and a discussion which considers only part of the results make a further revision necessary. I will try to explain my concerns in detail below.

**Detailed comments:**

(line numbers apply to the Response to Reviewer Comments in bg-2020-15-AC2-supplement.pdf).

L45: I am not sure about the policy of Biogeosciences, but I think this map is basic information that belongs into the paper proper, not into a supplement

Done. We have moved Supplementary Material S1 (now Fig. 1) into the main manuscript.

L64: This is not sufficient. As a reviewer of the paper, but also as a reader I do not want (and usually do not have the time) to delve into any data repositories to search for information which is essential for the interpretation of the data, and I see no reason why the authors do not provide this information here.

Agreed. We have created an additional figure (Supplementary Material S1 A-B) accounting for this criticized aspect. Supplementary Material S1A shows which seawater samples were sampled in which year and should help to clarify on the temporal aspect. As we find samples from different years within the same near-bed water cluster (consider for example different coloured points in BW4, Fig. S1A), we conclude that time is not the main driver of similarities or dissimilarities in microbial community compositions in our study. We have added an according statement into the revised manuscript (LL219-222). Supplementary Material S1B depicts the locations of sponge sampling in relation to seawater sampling positions.

L69: see comment above. And the information that seawater and sponges were sampled in close vicinity should be provided in the Methods section - this would be just one additional sentence.

Agreed. See comment above, we have added a new figure as Supplementary Material S1B. This figure shows the locations of sponge sampling in combination with the seawater sampling positions. We have added a sentence into the manuscript (LL 126-127) saying that seawater and sponges were sampled in close vicinity, but not exactly at the same position. This aspect, that sponges and seawater were not always sampled exactly at the same position (despite close spatial proximity) is also the main reason why we performed and want to stick to our machine learning approach (Fig. 3 and Supplementary Material S1). For those sponges that were not sampled exactly at the same position as seawater samples, the machine learning approach helped to define the sponge sampling location's cluster affiliation. An according sentence has been added into the manuscript (LL261-264).

L111 ff: I am still not happy with these extrapolations. It is far from clear which parameters go into this "machine-learning process". For example, how were (isolated) BW1 regions outside the data points discriminated from BW2 in the same area - just from depth, which is the only parameter available outside the sampling points? This extrapolation is particularly strange since only one BW1 data point exists, and a possible Taylor column which may be responsible for the BW1 cluster, is locally restricted. Currently, it is kind of a magic black box - something goes in (but the reader does not know what) and the predictions come out. Certainly it is not necessary to go into details, but some basic information of what's going on in the black box would be very useful. And the purpose of the extrapolations/predictions is still not clear: For the outcome of the study, they are not necessary, and they are in fact not used and interpreted in the paper.

Done. A sentence has been added to the manuscript specifying the input of the machine learning approach (LL 184-186). Geographic locations (coordinates) and cluster affiliations of the 114 seawater samples have been used as input. The cluster affiliations were predicted into space by using a k-Nearest-Neighbor-algorithm approach. The main purpose for conducting this machine learning approach has been to allow predictions of seawater cluster affiliation at any geographic point across Schulz Bank. In our case predictions at the geographic points of the sponge sampling locations were the most relevant output of this approach. Please consider also our argumentation above about the attribution of sponge samples to respective seawater near-bed clusters.

L126: I fully agree that Fig. 3 provides a very good conceptual model of how the various parameters around Schulz Bank may interact. But I still feel that this figure does not belong into the results section, because it does not present results, which have not been shown before. In the text of the results section Fig. 3 is mentioned only in connection with the temperature profiles (which obviously are not own results and are hardly discernible in the figure), and that these determine the water mass distribution. Because the limits of the water masses are not indicated, even this information cannot be deduced from the figure. Nethertheless, the figure is certainly useful for the interpretation of the results, and as such it is in fact used in the Discussion, where it should be placed.

Done. We have moved this figure (now Fig. 7) and the respective text from the Results to the Discussion (LL330-335).

L150: This has to be clarified. According to M&M, 19 stations were sampled, and exactly these 19 stations show up as dots in Fig. 2B, one of which is labelled BW1. Have there multiple (independent) microbial samples been taken per station? This is not mentioned in M&M, but is important information.

We have sampled 19 stations, each with 2 sampling depths (mid-water and near-bed water). At each station and at each depth we took 3 biological replicates (from different Niskin bottles) for analyses of the seawater microbial community composition. This sums up to 114 seawater samples in total. Please consider our modified description in L158-159.

L163: I am sorry if the authors feel that my comment criticises their sampling design. I am fully aware of the limitations of deep-sea sampling, and I acknowledge that the authors collected an impressive dataset. My comment aimed at the incomplete information provided in the text - one example is shown in the comment above.

We have addressed all comments above and elaborated further on the manuscript text as requested by the reviewer (see above).

L204: Again, I do not deny the value of the study, on the contrary. But, just looking at the data, some patterns are quite striking, and even if they cannot be tested statistically, they may be relevant and should be considered, e.g. the relative abundances in BW1 being very different from the other clusters.

In this point we disagree with the reviewer. We think it would not be appropriate to draw conclusions such as that relative abundances at the seamount summit (BW1) are very different from the other parts of the seamount, as this pattern is not constant across the different sponge species (consider for example S. rosea and L. complicata data in Fig.6). We think that an over-interpretation of our data is not helpful in this respect. We have already mentioned in the previous revision round that we are confident that our study provides valuable insights into the trends across clusters, but that we think we do not have appropriate data to go for pair-wise assessments.

L214 (and elsewhere in the text and figures): the F value in ANOVA is a ratio, and hence the dF comprise two values.

Yes, that is right. We have modified Fig.4 (plus its figure legend) and added the second values (df2) into all sub-plots. In addition we have added all df2 values into the manuscript text (LL216-217, LL241, LL247, L248).

L240: It is fully ok to discuss the presented data, but since the stations are not directly comparable due to the different distances between near-bottom and water column samples, some cautious statement about the limitations of the conclusion would be appropriate

Done. We have added several sentences into the revised manuscript mentioning the limitations of our study in this respect (LL319-324).

L244: I do not know where this knowledge comes from, but it is simply wrong. Given the same driving forces, anticyclonic

Taylor caps/columns are generated also at seamounts in the southern hemisphere, but of course in counter-clockwise direction (anticyclonic = clockwise in northern hemisphere, counter-clockwise in southern hemisphere). Examples can be found, for instance, in Rogers et al. 2017, Pelagic communities of the South West Indian Ocean seamounts: R/V Dr Fridtjof Nansen Cruise 2009-410, DSR II Vol. 136

We have changed the respective sentence to: "This oceanographic phenomenon describes an isolated anti-cyclonic flow circulation pattern over a seamount and hence may promote temporary spatial isolation of a seamount ecosystem from adjacent waters." (L327-329).

L293: I cannot follow this argumentation ("… would enormously inflate the discussion"), and I find it strange to leave the interpretation of results to the reader. If results of the other clades and sponges do not add to the discussion or are not relevant, they can be omitted. The authors now state that Chloroflexi in G. hentscheli are discussed as "representative example". For what is this example representative? Does it mean that the results are similar in the other phyla and clades? Then this has to be stated. But looking at the results, this is certainly not the case.

As stated above we see the main value of our study in the description of general trends in microbial diversity across the determined near-bed water clusters. With Fig. 6 (and the whole network analysis behind it) we were particularly interested to address the following two questions in respect to sponge microbiomes: (i) Are the same microbial phyla significantly enriched or depleted in all three sponge species? (Question A); and (ii) Are only low abundant microbial phyla significantly enriched or depleted in all three sponge species? (Question B).

To answer these two questions we need to show all microbial phyla, but we see no value in discussing each and every phylum in detail. We have submitted this paper to Biogeosciences and not to a microbiology journal. We therefore think that most of the readers will also not be interested in a lengthy discussion of each and every microbial phylum.

Further, we disagree with the reviewer in his statement that we "leave the interpretation of results to the reader". Please consider our interpretation of the results in respect to Question A in LL425-428, and in respect to Question B in LL388-392.

However, we have realised just now thanks to the reviewer, that our formulation "representative example" is ambiguous. We have added a short explanation addressing this issue into the manuscript (LL414-418).

L324: see also my comment to L64 - the sampling design in relation to the three cruises is not comprehensible from the information given in the paper, and searching in Pangaea is not a reasonable option. And I still miss some convincing statement why interannual variability is no issue.

We have addressed this concern in our replies above. Briefly, we have added Supplementary Material S1 into the revised version of the manuscript. Further, we have modified parts of the manuscript text to accommodate for this request.

L330: This is exactly the point. If an influence (of biogeochemical factors) is suggested, the underlying assumption is a causal relationship, which my comment pointed at.

We agree with the reviewer that solely statistical correlations between biogeochemical parameters and microbial diversity, but no causal relationships can be established based on 16S data. We are fully aware that for causal relationships usage of other techniques, such as genomic inventories or stable isotope tracing, would be more appropriate. With this study we did not have the intention to describe causal relationships. As knowledge on deep-sea sponge microbiomes is very limited (less than a dozen studies have been published today), we have the opinion that statistical correlations provide an inevitable starting point before one can derive at causal relationships. We cannot deny that we had testing of a scientifically reasonable hypothesis (an assumption) in mind, when we set up this study.

Taking the reviewers remark seriously, we have modified the criticized sentence in the conclusion (LL433) to:

[revised manuscript text omitted]